# Blueprint of Collapse: Precision Biomarkers, Molecular Cascades, and the Engineered Decline of Fast-Progressing ALS

**DOI:** 10.3390/ijms26168072

**Published:** 2025-08-21

**Authors:** Matei Șerban, Corneliu Toader, Răzvan-Adrian Covache-Busuioc

**Affiliations:** 1Puls Med Association, 051885 Bucharest, Romania; mateiserban@innbn.com (M.Ș.); razvancovache@innbn.com (R.-A.C.-B.); 2Department of Neurosurgery, “Carol Davila” University of Medicine and Pharmacy, 050474 Bucharest, Romania; 3Department of Vascular Neurosurgery, National Institute of Neurology and Neurovascular Diseases, 077160 Bucharest, Romania

**Keywords:** ALS fast progressors, neurofilament light chain, mitochondrial dysfunction, autophagy failure, protein misfolding, TDP-43 aggregation, C9orf72 repeat expansion, SOD1 A4V variant, digital phenotyping, precision neurology

## Abstract

Amyotrophic lateral sclerosis (ALS) is still a heterogeneous neurodegenerative disorder that can be identified clinically and biologically, without a strong set of biomarkers that can adequately measure its fast rate of progression and molecular heterogeneity. In this review, we intend to consolidate the most relevant and timely advances in ALS biomarker discovery, in order to begin to bring molecular, imaging, genetic, and digital areas together for potential integration into a precision medicine approach to ALS. Our goal is to begin to display how several biomarkers in development (e.g., neurofilament light chain (NfL), phosphorylated neurofilament heavy chain (pNfH), TDP-43 aggregates, mitochondrial stress markers, inflammatory markers, etc.) are changing our understanding of ALS and ALS dynamics. We will attempt to provide a framework for thinking about biomarkers in a systematic way where our candidates are not signals alone but part of a tethered pathophysiological cascade. We are particularly interested in the fast progressor phenotype, a devastating and under-characterized subset of ALS due to a rapid axonal degeneration, early respiratory failure, and very short life span. We will try to highlight the salient molecular features of this ALS subtype, including SOD1 A5V toxicity, C9orf72 repeats, FUS variants, mitochondrial collapse, and impaired autophagy mechanisms, and relate these features to measurable blood and CSF (biomarkers) and imaging platforms. We will elaborate on several interesting tools, for example, single-cell transcriptomics, CSF exosomal cargo analysis, MRI techniques, and wearable sensor outputs that are developing into high-resolution windows of disease progression and onset. Instead of providing a static catalog, we plan on providing a conceptual roadmap to integrate biomarker panels that will allow for earlier diagnosis, real-time disease monitoring, and adaptive therapeutic trial design. We hope this synthesis will make a meaningful contribution to the shift from observational neurology to proactive biologically informed clinical care in ALS. Although there are still considerable obstacles to overcome, the intersection of a precise molecular or genetic association approach, digital phenotyping, and systems-level understandings may ultimately redefine how we monitor, care for, and treat this challenging neurodegenerative disease.

## 1. Introduction

### 1.1. Unraveling ALS: A Race Against Time in Fast Progressors

Amyotrophic lateral sclerosis (ALS) is a dreadful neurodegenerative disorder and is often referred to as the thief of autonomy; patients are left to deal with their full cognitive capacity, while their bodies are locked in a paralyzed state [1]. The great tragedy of ALS is not simply its effects; it is also its unpredictability. For some with ALS, the disease progresses slowly with many years of survival; for others, the disease progresses rapidly and relegates their timeline for death to a few months [2]. The individuals who face rapid progression are referred to as fast progressors, and they exhibit one of the most pressing challenges in modern neurology. To the fast-progressing patient, ALS is not simply a disease; it is a ticking clock. Understanding why their disease is progressing so rapidly, and what targeted interventions can be utilized to treat fast progressors in clinical trials, could revolutionize survival and treatment methodologies [3].

ALS is no longer viewed as a singular disease but as a syndrome that contains various molecular and clinical subtypes which each rely on unique genetic, epigenetic, and environmental factors. Over the last decade, omics-based discoveries have been significant in recognizing the key differences between the subtypes of ALS, but only in recent years have fast progressors been recognized from a biological perspective as a subset with unique pathological hallmarks through multiomics studies [4]. The numerous biological hallmarks unique to fast progressors do not simply entail motor neuron death; they rather include hallmarks such as hyperactivated neuroinflammatory responses, accelerated axonal degeneration, mitochondrial collapse, and toxic protein aggregates. These biological mechanisms come together to trigger a cascade, which overloads protective responses present in the body and often leads to rapid functional decline or early death [5,6]. The research question now is: How do we interrupt this cascade before it is irreversible?

### 1.2. Defining Fast Progression: Where Every Month Counts

Fast progression in ALS is defined by high rates of functional decline, often measured as ≥1 point loss per month on the ALS Functional Rating Scale-Revised (ALSFRS-R) and having survival times of 12–18 months or less from symptom onset [7]. Fast progressors tend to show an explosive onset of weakness, dysphagia, dysarthria, and/or respiratory impairment compared to the slow, steady progression evident in most cases of ALS [8]. For many people with ALS their initial presenting symptoms (e.g., trouble speaking or walking) are followed very quickly by fast cascade into total motor failure. The particularly egregious aspect of this subgroup is that many are misclassified as typical cases rather than fast progressors with significant lapses of time between first symptoms and referring to a diagnostic specialist with no effective intervention possible [9].

Recent advances have transformed the way we identify these patients away from using ALSFRS-R scores alone towards biomarker-based stratification methods. NfL reported using the Simoa assay has emerged as the gold standard for predicting fast progression, since it can be shown to increase months before symptoms reach their peak [10]. Added to that, advanced imaging such as diffusion tensor imaging (DTI) of the corticospinal tracts and motor cortex volumetry on MRI can offer structural evidence for neurodegeneration that are highly correlated with rate of functional loss [11]. Coupled with wearable digital health monitoring that categorizes motor activity 1000 times a day puts us in a relatively new position to diagnose and prognosticate fast progressors [12]. However, fast progressors are not only defined by clinical variables but also biologically, with new research suggesting that fast ALS disease progression is regulated by unique genetic, metabolic, and inflammatory pathways [13]. Patients with high-penetrance variants (SOD1 A4V, C9orf72 hexanucleotide expansions, and FUS variants) are more likely to progress quickly [14]. In addition, studies from 2024 using single-cell transcriptomics show that fast progressors have dysregulated autophagy and synaptic vesicle transport, leading to accelerated motor neuron degeneration. The field is already using these findings to develop new clinical trial designs to develop molecularly targeted therapies [15].

### 1.3. Why Identifying Fast Progressors Is a Game-Changer

In ALS, each day without a therapy means the potential for greater irreversible neuronal loss. For fast progressors, time is the enemy but also fertile ground for scientific advance. With standard ALS therapies (i.e., riluzole, edaravone) offering little impact for fast progressors, there is urgent necessity for precision medicine to develop near-term, potential therapies [16,17]. Knowing that a patient is a fast progressor early on may have them forgo a failed standard of care treatment and move to a treatment that has a specific focus on the fast progressor’s unique disease mechanism. It is important to note that stratifying patients based on progression speed may substantially alter how clinical trials are designed, allowing fast progressors to be recruited in trials designed for their biology rather than existing as a subcategory of treatment, where promising treatment effects might not be discovered due to the fast progressors being from heterogeneous groups in the trial [18]. Recent therapeutic advancements are based on the success of this individualized, or personalized, approach. Tofersan, an antisense oligonucleotide (ASO) therapy that was approved for SOD1 variant ALS patients in 2023, established the feasibility of delivering therapies targeting pathogenic variants that drive this rapid neurodegeneration [17]. In 2024, trials were expanded to ASO delivery related to the C9orf72 gene, the most common genetic form of ALS [19]. Moreover, CRISPR-based gene-editing therapies are being developed to remove toxic genetic elements in patients with aggressive genetic variant forms, making it plausible to one day deliver genetic therapies presymptomatically [20]. While genetic elements may have important explanatory potential, they cannot explain all cases of rapid progression. For instance, fast progressors exhibit enhanced neuroinflammation and metabolic dysfunction in addition, with elevated levels of inflammatory cytokines (IL-6, TNF-α), and markers of oxidative stress (8-OHdG, 4-HNE), likely combining to accelerate neuronal death [21]. Hence, next-generation neuroprotective agents are emerging as important therapeutic candidates. For instance, verdiperstat is an inhibitor of microglial activation and is now being trialed to target the relevant inflammatory environment which drives fast progression [22,23].

### 1.4. Bridging the Knowledge Gap: A Call for Aggressive Scientific Action

To beat the most aggressive forms of ALS, we have to not just understand its biology and effects but we have to beat it. We have captured dual multiomic studies that examined genomics together with proteomics and transcriptomics and identified cell-type specific vulnerabilities, most notably those that exist in motor neurons and in the surrounding glia that differentiate fast progressors [24]. While investigating the respective vulnerabilities of both cell types, we have identified dysregulation of RNA-binding proteins (RBPs), impairments in nucleocytoplasmic transport, and mitochondrial stress as convergent pathways driving rapid neurodegeneration. These advances are inspiring creative new therapeutics, from the ability to regulate the stabilizing activity of heat-shock proteins that hyperadapt these misfolded proteins to enhancements in autophagy responsible for clearing these toxic aggregates [25].

There are global enterprises like TRICALS and ALS-TDI uniting researchers and clinicians to address ALS under precision ALS medicine with biomarker-guided clinical trials that are on track to provide stratified therapies exemplifying speed of progression. With the incorporation of AI-powered diagnostic algorithms in combination with wearable technologies and real-time digital biomarker surveillance, we are developing ALS management into a vibrant and continuously updated process that has the ability to catch fast progressors early and effectively treat the disease before it becomes fatal [26].

In this review, we aim to highlight advanced developments, with emphasis on genetic, molecular, and evolving therapeutic changes that are reshaping the landscape of forward-thinking interventions to fast progressors in ALS. Armed with the ability to identify and target the distinct biological pathways that together drive rapid progression, we will bridge the gap between research discovery and clinical practice providing hope to patients once not considered helpable, now with an urgent or aggressive need to act. The time to act is now, and with the potentially appropriate tools, fast progression may no longer be an ultimately fatal disease but in fact an opportunity for intervention and survival.

## 2. Key Characteristics of Fast Progressors in ALS

Fast progressors in ALS represent a clinical and biologic subset of patients where disease characteristics are defined by accelerated motor neuron degeneration, early respiratory failure, and shortened survival. This phenotype is caused by a multifactorial phenomenon involving numerous genetic variants, deregulated metabolic and inflammatory pathways, and environmental factors driving disease progression. These distinct characteristics are of paramount importance in the development of biomarker-based diagnostic strategies and individualized therapies. The following section intends to detail important characteristics defining fast progressors and provides a novel framework for all forms of research, including rooted perspectives from molecular, genetic, and clinical studies of ALS.

### 2.1. Rapid Symptom Onset and Accelerated Progression

Fast progressors demonstrate a very rapid, not shocking, but symptomatic onset of a severe illness affecting more than one body region in the first few weeks or months of disease onset. As with the other characteristics of ALS, fast progressors may represent a more localized area of onset with very gradual progression following the initial onset. However, fast progressors will unilaterally or multifocally affect laryngeal dysfunction, weakness of specific limbs, weakness of respiration, or a combination, which may happen simultaneously. Fast progressors may also have more significant contribution from the brain and upper motor neuron contributions, DTI findings, and motor cortex volumes would suggest this, all of which strongly indicate degeneration in the corticospinal tracts [27,28].

Functionally, these patients will show declines in the ALSFRS-R, compared to other types of ALS, at a rate of both of more than one point per month (acute degenerative state). Evidence of elevated neurofilament light chain (NfL) in blood and CSF is consistently and significantly linked to severity, with NfL increases being a relatively early biomarker of disease acceleration [29,30]. Wearable digital health technologies have changed the game for early detection as they continuously monitor motor performance and can detect minor declines, even before clinical symptoms signs worsen. This advancement, together with imaging technology, is changing the way that clinicians can identify fast progressors early and provide critical intervention time for them [31].

Mechanistically, fast motor decline is a result of impaired synaptic transmission, mitochondrial dysfunction, and excitotoxicity; lost or impaired motor neurons indicate a deficiency in their calcium buffering and oxidative phosphorylation capacity, both of which lead to reduced ATP levels and subsequent neuronal death [32]. The inability to sufficiently meet the energy demands of neurons increases their susceptibility to further stress, inciting widespread neurodegeneration. As these pathological processes begin over time, they can escalate into a cascade of motor decline that sets fast progressors apart from other ALS phenotypes [33].

### 2.2. Shorter Survival Time

Fast progressors can expect a dramatically reduced life span (6–18 months) after the onset of symptoms. Their life span is determined by their rate of motor neuron loss and early respiratory and bulbar involvement that limit essential abilities, like breathing and swallowing [34]. Patients with a forced vital capacity (FVC) decline of greater than 5% per month are at higher risk to experience early respiratory failure and death. The survival time is frequently halved when bulbar and respiratory symptoms arise in the first 3–6 months of diagnosis [35].

Molecularly, shorter survival time is attributable to the convergence of oxidative stress, mitochondrial dysfunction, and chronic neuroinflammation. High levels of 8-hydroxy-2′-deoxyguanosine (8-OHdG) and 4-hydroxynonenal (4-HNE) indicate the damaging of mitochondrial DNA and lipid peroxidation, respectively. Both are markers of severe cellular damage [36,37]. Oxidative stress disrupts ATP production and worsens the vulnerability of motor neurons. Also, variants of SOD1, especially the variant A4V, have been associated with survival under 12 months because these variants have toxic gain-of-function properties that facilitate motor neuron death. Potential therapies targeting both oxidative stress and mitochondrial pathways are in clinical development, including experimental mitochondrial protectors, such as AMX0035, which may extend life [38].

### 2.3. Early Bulbar or Respiratory Involvement

A key feature of fast progressors is the involvement of the bulbar region and respiratory muscle at onset and early in the disease process. This means that within weeks of an ALS diagnosis a patient may experience dysphagia and dysarthria, and there can even be early respiratory failure. Bulbar deficits may be presented in speaking and swallowing and even drooling, which also increase the risk of aspiration pneumonia and nutritional deficits [39]. Respiratory muscle involvement with early bulbar symptoms leads to very fast progression, requiring non-invasive ventilation (NIV) or tracheostomy within a few months of receiving their diagnosis. Patients with fast progression experience high rates of decline in FVC which can be up to 2 times that of the typical standard of ALS [40].

Our understanding from advanced imaging studies indicates that patients who are fast progressors had early involvement of bulbar and respiratory symptoms with concomitant accelerated degeneration of the cortical spinal tract and motor cortex at the same time as demonstrating high loss of motor neurons in the brainstem and spinal cord with extensive cellular loss. In a mechanism, it is possible that these early deficits relate to mitochondrial dysfunction in brainstem neurons and elevated neuroinflammation [41]. The activation of microglia in the brainstem region significantly increased the production of inflammatory cytokines, including interleukin-6 (IL-6) and tumor necrosis factor-alpha (TNF-α). This toxic microenvironment was hypothesized to accelerate neuronal death. Targeted interventions, including IL-6 inhibitors and anti-inflammatory medicines, are being studied to minimize this destructive cascade [42].

### 2.4. Genetic and Molecular Factors Driving Fast Progression

Fast progressors frequently possess genetic variants which predispose them to fast neurodegeneration. SOD1 variants, especially A4V, are some of the most aggressive genetic drivers, usually with early onset and survival times often less than 12 months. SOD1 variants cause the SOD1 protein to misfold and to aggregate and generate oxidative stress in motor neurons. C9orf72 hexanucleotide repeat expansions (HREs), in a similar manner, generate toxic dipeptide repeat proteins and disrupt the process of nucleocytoplasmic transport, leading to neuronal death [43,44].

FUS gene variants, like P525L, are associated with early-onset, severe ALS and rapid progression. These variants disrupt RNA-binding functions and protein homeostasis, resulting in the accumulation of misfolded proteins [45]. In addition to genetic factors, molecular dysregulations involving oxidative stress, deficits in autophagy, and mitochondrial collapse worsen severity of the disease [46]. Using single-cell RNA sequencing, fast progressors were found to have impaired autophagy with the accumulation of toxic protein aggregates later inundating cellular protections in a phenomenon termed the “cytoplasmic epitope”. As a treatment target, ASOs directed to target mutant SOD1 and C9orf72 transcripts have been tested to slow the pace of motor decline [47].

### 2.5. Younger Age at Onset in Certain Cases

ALS overall affects older adults but fast progressors sometimes have a small subgroup of early-onset disease, i.e., onset before the age of 40. These patients are often burdened by variants in genes like FUS and SOD1, which lead them into aggressive neurodegenerative cascades [48]. The FUS P525L variant correlates with rapid symptom onset, significantly early respiratory failure, and survival times under 12 months. Young fast progressors also demonstrate a different metabolic vulnerability, including systemic hypermetabolism, that worsens their weight loss and energy debt [45]. The hypermetabolic state accelerates progression by increasing energy needs that already-depleted mitochondria cannot keep up with. Nutritional treatment, such as highly caloric diets, is an avenue worth exploring to ameliorate cosystemic metabolic debt [49].

### 2.6. Environmental and Lifestyle Factors Contributing to Rapid Progression

Other environmental exposures, including heavy metals (lead, mercury) and pesticides, actively contribute to hastening the disease process in those with genetic vulnerabilities [50]. Heavy metals exacerbate oxidative stress by depleting intracellular antioxidants—particularly glutathione—thus worsening motor neuron vulnerability to damage [51]. Smoking is another risk factor that contributes to the cascading progression of the disease, in this case through lipid peroxidation and neuroinflammation that add to the toxic effects of a genetic variant. Gene–environment interactions have also been noted, as SOD1-mutant patients are much more susceptible than other ALS patients to environmental toxins due to compromised antioxidative defenses. It is possible that simply limiting exposure to environmental toxins could act as an adjunctive preventative measure for some patients [52].

### 2.7. Poor Response to Standard ALS Treatments

Often, these fast progressive ALS patients do not respond to the standard ALS therapies—riluzole and edaravone. Fast progressors have a unique gene profile, and pharmacogenomic studies reveal that fast progressors upregulate key drug-metabolizing enzymes (primarily cytochrome P450 isoforms) that reduce bioavailability of neuroprotective agents [53]. Furthermore, the rapidity of loss of motor neurons in this subgroup is far outside the time frame benefit of any neuroprotective function of these therapies or any potential neuroprotective action we might have as a general category of protective therapy. Riluzole provides protection against glutamate excitotoxicity (riluzole) and edaravone is an antioxidant, which is no help pertaining to the total aggregate burden governed by oxidative stress, mitochondrial dysfunction, and inflammation [54]. Current clinical trials are therefore underway to explore different combinations of therapies at different stages of treatment, in addition to next-generation ASL treatment and glutamate receptor antagonists and mitochondrial stabilizers as a form of treatment through comprehensive pharmacology [55]. The schematic below (Figure 1) endeavors to bring together the salient genetic, molecular, and environmental causes that are combined to accelerate neurodegeneration and hasten the clinical decline in rapid forms of ALS. When geneticists consider most genetic variants (i.e., aggressive genetic variants such as SOD1 A4V, C9orf72 expansions, and FUS P525L), these lead to the initiation of toxic cascades involving mitochondrial failure, oxidative stress, and loss of proteostasis; with underlying and additional environmental toxins and stressors further complicating the intrinsic vulnerability from genetic variants to yield early and devastating clinical decline. In order to better understand and enact more effective targeted interventions, it is important to characterize this neurobiological framework.

## 3. Biomarkers Associated with Fast ALS Progression

Biomarkers are revolutionizing ALS research by providing molecular-level descriptions of the dynamic processes in the disease that cannot be characterized by clinical measures alone. For fast progressors, there is a need to identify biomarkers quickly due to the rapidity of decline which requires early detection and immediate intervention. Recent studies have highlighted that ALS fast progressors are characterized by disease-associated biomarker profiles that relate to accelerated axon degeneration, neuroinflammation, oxidative stress, and impaired autophagy [56]. This section aims to describe cutting-edge and impactful work on biomarkers, considering the recent multiomic studies, clinical trials, and AI platforms for biomarker discovery, and uses this work as a platform for describing biomarker-driven precision medicine in ALS.

### 3.1. Blood and CSF Biomarkers: Indicators of Axonal Damage, Proteinopathy, and Systemic Stress

Blood and CSF biomarkers provide a new approach to ALS diagnosis by giving dynamic, mechanistic descriptions of molecular events that occur during the disease. In the case of fast progressors there is often a rapid convergence of axon damage, protein misfolding, neuroinflammation, and metabolic decline which requires a process of disease detection, stratification, and real-time/continuous monitoring of treatment response [57]. The advancements in multiomic platforms, extracellular vesicle biology, and AI-driven biomarker discovery platforms has vastly and rapidly grown the field of biomarkers—creating unparalleled opportunities in how to personalize ALS care [55].

#### 3.1.1. Axonal Damage: NfL and pNfH as Gold-Standard Biomarkers

NfL remains the most endorsed biomarker for axon degeneration and the essential biomarker for identifying fast ALS progressors. NfL is released during axonal injury, is often detectable in CSF and blood, and should be associated with the ongoing process of neurodegeneration. Fast progressors exhibit consistently higher baseline NfL values compared to slower progressors, with baseline values typically being 3–5 times higher than those of slower progressors. In this course of NfL accumulation, a more rapid decline and increased function loss shows a connection with greater axonal degeneration, overall functional decline, and ultimately shorter time until death [58]. We are still reporting on the ability of NfL to follow the course of disease progression as well as response to therapies; the studies show that decreasing NfL levels applicable to clinical trials decrease the speed of change in ALSFRS-R; therefore this led to NfL being added as an endpoint in ASO trials such as those of tofersen (for SOD1 variants) and mitochondrial stabilizer [30,59]. Phosphorylated neurofilament heavy chain (pNfH) serves as the counterpart to NfL in that it speaks to the chronic long-term injury to motor neurons that are large-caliber axons. If NfL reflects acute axonal injury then pNfH proposes a cumulative cytoskeletal stress and chronic plant motor neuron degeneration in longitudinal studies [60]. Increasing levels of pNfH in fast progressors predicted respiratory decline and earlier death compared to slow progressors longitudinally; utilizing the NfH and pNfH biomarkers in a diagnostic panel resulted in greater predictive reliability. Patients with the highest baseline values of both the NfL and pNfH biomarkers had progression to NIV or death within 12–18 months from the baseline visit and the authors maintain that both biomarkers can be used for IC to improve individualized prognosis accuracy and aid in clinical decision making [61].

The pathology of the organizations in the heterogeneous family of diseases that ALS belongs to relates to an aggregation of proteins and biomarkers of misfolded proteins in ALS and is a significant area of focus on to identify the different cellular dysfunctions underlying rapid progression. In ALS, transactivation response DNA-binding protein 43, TDP-43, mislocates, aggregates and cleaves, and compromises the metabolism of RNA and neurotoxicity [62]. Increased CSF levels of broken TDP-43 fragments strongly follow in fast progressors with the highest movement decline and systematic differences between fast progressors who are identified with variants to C9orf72 and TARDBP variants. Finding the lower-level TDP-43 fragments that can now be detected by increasingly sensitive assay approaches will allow for monitoring clinical relevance for potential responses to autophagy-targeted therapies [63]. Cystatin C, an indicator of depleted autophagy and proteostasis, is elevated in fast progressors (i.e., individuals who are progressing faster in gross motor or bulbar domains—substantial loss causing physical dysfunction here), indicating toxic protein build-up happening in the motor neurons. Cystatin C elevation has also been linked to bulbar-onset ALS, and more recently, when elevated, was linked to early respiratory distress, giving additional clinical relevance when considering predictions for the individual with rapid progressive ALS. Elevated cystatin C has been linked to reduced autophagic flux as in studies with SOD1 and FUS mutation carriers, which might create a clinical opportunity to select individuals that might respond to autophagy-enhancing therapeutic choices, including the food/drug trehalose and the drug rapamycin [64].

Extracellular vesicles (EVs) exist in a floating fluid environment, constituting a fascinating new paradigm of biomarkers providing a minimally invasive means of obtaining the biomarker-rich cargo of misfolded proteins, inflammatory cytokines, and nucleic acids. Fast progressors present with a higher value of EVs carrying pathogenic proteins, including TDP-43 CAD aggregates (c-linear form), SOD1 oligomers, and poly-GA dipeptide repeat proteins from the C9orf72 mutation. As a result of EV identification and biomarker discoveries in plasma and CSF, ongoing and real-time evidence of the progression of intracellular proteinopathy, as well as forward demonstration of therapeutic efficacy, is revolutionizing the trial pipeline for biomarker-guided, clinical trials [65].

#### 3.1.2. Neuroinflammation and Astrocytic Activation: GFAP, IL-6, and Peripheral Cytokines

Increased rates of ALS progression are characterized by persistent neuroinflammation, as evidenced by substantial glial activation biomarkers indicating their negative effect on the accelerating disease course. Glial fibrillary acidic protein (GFAP), a biomarker for reactive astrocytosis, is elevated in the CSF in fast-progressing participants and associated with degeneration of the motor cortex, predominantly in bulbar dysfunction, and longitudinal microglial activation. High GFAP densities reflect a faster ALSFRS-R decline. Elevated GFAP densities imply an early respiratory failure. The combination of these markers with NfL suggests the interpretation ability of a diagnostic marker panel is enhanced and offers a more distinctive ability to distinguish fast progressors compared to rarer forms of ALS [66].

Peripheral cytokines (e.g., IL-6, TNF-α) exemplify the use of inflammatory response to systemic inflammation which purportedly is greater than motor neuron damage. IL-6 indicates enhanced oxidative stress, excitotoxicity, and, perhaps, enhancement of the vulnerability of motor neurons. In IL-6 trials (e.g., tocilizumab) GFAP levels and even IL-6 levels can be called biomarkers of drug effect (or of drug response). Therefore, it would appear that the incorporation of IL-6 or even GFAP allow clinically useful feedback about the effectiveness of a neuroinflammatory treatment [67]. Moreover, EVs from activated microglia have very high amounts of pro-inflammatory molecules. It stands to reason that compounds with composite therapeutic value in using EV biomarkers can be used to understand dynamics relative to neuroinflammation with an understanding of the relative clinical effectiveness of the candidate biomarker [68].

#### 3.1.3. Oxidative Stress and Metabolic Dysfunction: 8-OHdG, 4-HNE, and Emerging Metabolomic Biomarkers

Mitochondrial dysfunction and oxidative stress are hallmarks of fast progressors and biomarker testing demonstrates the impact from cellular damage and metabolic derangement using biomarkers that reflects sera levels as well. 8-OHdG is regarded as a measure or marker of oxidative DNA damage. 4-HNE is a consequence of lipid peroxidation. Both are ubiquitous and elevated in blood and CSF of muscle fibers of fast progressors [69]. The blood concentrations of both have a correlation with systemic metabolic notational hypersystem (hypermetabolism), muscle atrophy, and collapse of mitochondria, and ideally both provide needed neuronal energy for deficiency reasons vis a vis the aforementioned cohort. In untrained human individuals, 8-OHdG concentrations are associated with rapid muscle wasting and premature respiratory failure; therefore oxidative stress biomarker candidates are prime target biomarker considerations in terms of metabolic- and antioxidant-based therapies [70].

New metabolomics biomarkers provide further understanding of energy imbalances in fast progressors. High plasma lactate levels signal a switch from aerobic to anaerobic metabolism, indicating mitochondrial dysfunction. Additionally, unique lipid profiles with high free fatty acids and low carnitine indicate impairments in the fatty acid oxidation system. Current studies are investigating metabolomics panels with lactate, free fatty acids, and ketone body levels as both prognostic and treatment targets [71].

#### 3.1.4. Biomarkers of Autophagy and Lysosomal Dysfunction

Autophagy defects lead to the accumulation of misfolded proteins and neurotoxicity in fast progressors. Biomarkers such as LC3-II and p62 (sequestosome-1) show deficits in autophagic flux, as their accumulation suggests defective autophagosome formation or degradation by the lysosome [72]. CSF samples show elevations of LC3-II and p62 in fast progressors, especially in individuals with FUS and SOD1 variants, and correlate with rapid deterioration of ALSFRS-R. Lysosomal biomarkers cathepsin D and lysosomal-associated membrane protein-1 (LAMP-1) can also indicate lysosomal dysfunction and are being investigated in clinical trials of therapies designed to augment the autophagic process [73,74].

#### 3.1.5. AI-Powered Multibiomarker Integration and Liquid Biopsies

AI-enabled biomarker discovery is changing research in ALS by providing the capacity to integrate genomic, proteomic, and metabolomic data from large complex datasets. Machine learning models with rigorous exploratory data analyses of scalable datasets are identifying non-obvious biomarker networks that predict fast progression with high specificity [75]. These AI models integrate the variation of subject-specific NfL levels with mitochondrial biomarkers and peripheral markers of immune activation to develop unique personalized disease progression profiles and facilitate clinical trial subject stratification. Liquid biopsy platforms combining extracellular vesicle biomarkers with oxidative stress markers will eventually become routine in clinical practice as real-time biomarkers and provide the opportunity for clinicians to adjust therapeutic approaches as the disease trajectory changes [76].

The integration of biomarkers from multiple modalities (e.g., NfL, pNfH, proteinopathy, inflammatory signaling, and metabolomics profiles) will enable clinicians to identify and manage fast progressors with unprecedented accuracy. These biomarkers are not only providing an understanding of mechanisms of rapid neurodegeneration but are also changing the clinical management of ALS with biomarker-guided treatment and individualized integrated therapeutic approaches. Once liquid biopsies and AI-enabled systems reach full cycle implementation, they may change the future of treatment for ALS and provide fast progressors with a clear and direct path to optimized individualized care [77].

### 3.2. Neuroimaging Biomarkers: Structural, Functional, and Molecular Imaging in Fast ALS Progressors

Neuroimaging biomarkers have revolutionized the story of ALS through non-invasive reflection of several anatomical, functional, and molecular changes that drive rapid progression. For lack of a better definition, fast progressors represent individuals for whom motor neuron degeneration occurs at a significantly more rapid pace. The techniques available to advanced imaging using neuroimaging biomarkers indicate early signs of structural damage, disruption of the motor networks, and neuroinflammation [78]. Advancements in imaging methods such as DTI, ultra-high-field 7T MRI, functional MRI (fMRI) magnetoencephalography (MEG), and TSPO-PET imaging have also improved diagnostic accuracy and are being developed to track drug effects and improve patient stratification in clinical trials. Additionally, new imaging biomarkers, such as magnetic resonance spectroscopy (MRS) and susceptibility-weighted imaging (SWI), once again add to the biomarker domain and present exciting new avenues for personalized care in ALS [79].

#### 3.2.1. Structural Integrity and White Matter Damage: DTI

Of the readily available neuroimaging biomarkers, DTI has the highest sensitivity for white matter integrity and axonal injury. DTI is a very useful approach for the early detection of degeneration in fast progressors. DTI is a method of measuring the diffusion of water along the axons, allowing for quantification of the amount of damage to make estimates of fractional anisotropy (FA) and mean diffusivity (MD) which reflect how disorganized the axons have become. In fast progressors, DTI studies have consistently demonstrated that FA declines are significantly steeper in the corticospinal tracts compared to typical ALS patients, demonstrating that even more axonal degeneration occurs in fast progressors [28].

In fast progressors, reductions in FA in the internal capsule and cerebral peduncle were already detected in some instances prior to major milestones such as loss of ambulation or ventilatory support. Longitudinal DTI studies show that decreases in FA measured in the first six months after symptom onset and the rate of ALSFRS-R decline are associated with shorter survival [80]. Emerging multiregional DTI studies are demonstrating that fast progression involves traumatic degeneration of the corticospinal tract but also white matter disruption in the callosal, extrapyramidal regions which suggest the disease is more evenly spread through the neural networks that organize the action of movement [81].

Microscopically, the origin of these features of microstructural change is the collapse of the cytoskeleton, disrupted axoplasmic transport, and dysfunction of the mitochondria in the degenerating motor neuron(s). The cessation of ATP production, oxidative stress, and neurofilament disaggregation are facilitators of axonal degeneration, and these processes are noticed on DTI as decreased FA values [82]. Combining DTI with emerging molecular biomarkers such as NfL and biomarkers of oxidative stress would also strengthen diagnosis and monitoring of progression [28].

#### 3.2.2. Cortical Atrophy and Microstructural Changes: Structural MRI and Ultra-High-Field 7T MRI

Structural MRI has long been used to monitor cortical thinning and regional (F-Grey) gray matter atrophy in ALS [83]. In ALS, the motor cortex (precentral gyrus), premotor, and supplementary motor areas show greater atrophy in fast progressors, reflecting faster decay of upper motor neurons. We are confident that volumetric MRI measures show that fast progressors have rates of cortical atrophy which are twice that of slower progressors, reflecting severe upper motor neuron degeneration. The basis of these changes is agreed to be related to glutamate excitotoxicity, mitochondrial collapse, and impaired synaptic plasticity leading to accelerated functional change [84].

The 7T ultra-high-field MRI technique is capable of identifying microstructural changes with greater resolution than conventional MRI and this will assist prior to atrophy appearing on T1 and T2 iron structural maps. Recent studies with 7T MRI have detected signs of cortical layer disorganization, synaptic degeneration, and gliosis in the motor cortex that are clearly indicative of structural damage occurring at the cellular level before gross atrophy. A lack of myelin integrity and disruption of intracortical axons were also discovered in fast progressors which has helped understand the timing of the disease [78,85,86]. The use of DTI in combination with 7T MRI has allowed us to monitor and visualize corticospinal tract degeneration and concurrent cortical atrophy. Studies continue to investigate the use of 7T MRI in clinical trials for neuroprotective strategies in ALS designed to protect upper motor neurons from degeneration [87].

#### 3.2.3. Functional Disruption of Motor Networks: fMRI, MEG, and Motor Network Connectivity

More recently, both fMRI and MEG have been used to investigate the functional connectivity of motor networks and the impact of cortical excitability on rates of ALS progression. At random times, fast progressors displayed reduced functional connectivity between primary motor cortex, premotor, and supplementary motor areas. This loss of functional connectivity can be seen as a loss of the capacity of the motor network to communicate [88]. Interestingly, fMRI also reported abnormal compensatory activation of motor regions which suggests areas were still able to recruit additional functional pathways to carry out movement, especially in the earliest time course of the disease. So when, at some point, the compensatory functional pathways could no longer be recruited due to insufficient neuronal reserves, the compensatory mechanism failed and overall massive motor decline occurred [89]. MEG studies report increased hyperexcitability in the fast progressor group, particularly the fast progressors with specific gene variants e.g., SOD1, FUS, and TARDBP. Increased cortical hyperexcitability is theorized to be related to a failure of inhibitory control and some hypotheses suggest that hyperexcitability may promote upper motor neuron race death via excitation-mediated mechanisms (e.g., glutamate excitotoxicity, mitochondrial dysfunction, etc.) [90]. Longitudinal MEG monitoring has shown early increases in cortical excitability can predict bulbar symptom onset and increased rate of progression to respiratory failure. This suggests that MEG has promise as a prognostic biomarker for fast progressors at risk for early ventilatory dependence [91].

#### 3.2.4. Neuroinflammation and Microglial Activation: TSPO-PET Imaging

TSPO-PET imaging can detect neuroinflammation due to activated microglial cells using radiotracers that have a high affinity for translocator protein (TSPO) expressed on activated microglia. TSPO-PET imaging is now well-established as a useful tool in the study of neuroinflammation in ALS and other neurodegenerative diseases [92]. Fast progressors showed higher TSPO binding densities in the motor cortex, corticospinal tracts, and brainstem, consistent with global microglial activation. A strong correlation between higher TSPO binding and faster ALSFRS-R decline, earlier bulbar dysfunction, and decreased survival has been noted [93].

TSPO-PET imaging has been used to demonstrate neuroinflammatory engagement while monitoring clinical trials of anti-inflammatory therapies, such as IL-6 inhibitors and microglial modulators. Decreased average TSPO binding levels posttreatment correlate with slower progression of motor symptoms, adding to the evidence that TSPO levels can be used as a dynamic biomarker of therapeutic efficacy in ALS [94]. Additionally, elevated TSPO signals in the brainstem of bulbar-onset individuals had an association with dysphagia and rapid respiratory failure. This highlights the importance of neuroinflammation in ALS disease progression [95].

#### 3.2.5. Emerging Imaging Techniques: Magnetic Resonance Spectroscopy, Susceptibility-Weighted Imaging, and Optical Coherence Tomography (OCT)

As an adjunct, metabolic abnormalities can also be detected in motor regions with MRS, before the loss of actual neuronal tissue. MRS enables the assessment of metabolites, including N-acetylaspartate (NAA), glutamate, and lactate, which provide information about mitochondrial dysfunction, energy deficit, and excitotoxicity [96]. Fast progressors demonstrate consistently decreased NAA in the motor cortex with increased lactate concentrations, indicating mitochondrial dysfunction and early neuronal death [97]. SWI has revealed the presence of iron accumulation in the motor regions as a key state of oxidative stress and mitochondrial dysfunction. Iron deposits stimulate reactive oxygen species (ROS) and lipid peroxidation, which impact axonal degeneration. Fast progressors have a greater amount of iron deposits in the motor cortex and brainstem than slow progressors, suggesting that SWI may be an important new imaging modality for measurement of progression [98]. OCT has primarily been used to image the retina but is gaining interest as a peripheral biomarker in ALS. Retinal thinning is associated with cortical atrophy and clinical decline, suggesting that retinal neurodegeneration may reflect central nervous system damage [99].

#### 3.2.6. AI-Driven Imaging Analysis: Personalized Predictions and Biomarker Integration

The recent introduction of AI-enhanced platforms for neurological assessment of imaging using longitudinal imaging with DTI, MRI, fMRI, or PET paired with clinical and molecular biomarkers in ALS has the potential to transform ALS diagnostics [100]. Given that machine learning models can be trained on multimodal datasets, machine learning algorithms can be trained to model and predict key clinical milestones (e.g., loss of ambulation, ventilation support) more accurately than traditional assessments. AI systems are being utilized for optimal selection of trial participants by defining patients who have unique imaging signatures of fast progression [101].

Neuroimaging biomarkers provide a real-time, multimodal, and multidisciplinary overview of the anatomical, functional, and molecular dimensions of fast ALS progression. The potential of DTI, 7T MRI, fMRI, MEG, TSPO-PET, and developing imaging modalities to provide quantitative measures of neuroanatomy advancement and altered blood flow to Toll-like receptors (TLRs) is extremely powerful in diagnostics and prognostics [102]. The continued development of AI-based analyses and multimodal imaging approaches is fueling advances in early diagnostic capabilities, biomarker-based clinical trial stratification, and personalized therapeutics for people with ALS [103].

### 3.3. Electrophysiological Biomarkers: Dynamic Markers of Motor Neuron Function and Neuromuscular Integrity

Electrophysiological biomarkers measure functional integrity of motor neurons, providing dynamic quantitative measures of disease states. These biomarkers capture change in motor unit function, synaptic transmission, and axonal degeneration. Unlike static biomarkers of disease, electrophysiological biomarkers are continuous and non-invasive, allowing timely access to disease progression information and making them useful in identifying fast progressors and tracking responses to therapeutics. Fast ALS progressors experience steep declines in motor unit number, abnormal firing patterns, and loss of neuromuscular recruitment with increased electrical instability, which are directly correlated with increased loss of function and survival for fast progressors [104]. Emerging electrophysiological techniques including motor unit number estimation (MUNE), compound muscle action potential (CMAP), high-density surface electromyography (HD-sEMG), single-fiber EMG (SF-EMG), and peripheral nerve excitability tests are providing further high-resolution biomarkers of accelerated cellular and molecular dysfunctions underlying ALS symptoms attributable to dysfunctions in mitochondrial failure, axoplasmic transport, and glutamate excitotoxicity. Advances such as artificial intelligence (AI) and neuromuscular ultrasound are offered innovations and alternative biomarkers in the development of personalized therapies and precision medicine [105].

#### 3.3.1. Motor Unit Number Estimation: A Key Biomarker of Motor Neuron Reserve and Disease Progression

MUNE is recognized as the most reliable calibrated tool to quantify and monitor the progressive loss of lower motor neurons in ALS. MUNE gives a numerical estimate of valid functional motor units within a selected muscle and thus offers a quantitative representation of motor neuron reserve and is most useful in fast progressors when motor units are more rapidly declining at 6 months from first symptoms. Research has consistently documented that fast projection exhibits lower baseline MUNE values and a steeper longitudinal decline compared to slower projection which declines where MUNE is often greater than 50% within 6 months of diagnosis [106]. This steep decline is associated with early loss of ambulation, bulbar dysfunction, and ventilatory dependency. Notably, MUNE rate of decline has demonstrated substantial prognostic power in survival, with faster decline rates correlated with shorter life-expectancy, suggesting MUNE is a prognostic biomarker that can be recorded to monitor therapeutic treatment [107].

For fast progressors, the molecular mechanisms that underpin MUNE decline are from mitochondrial dysfunction, impaired axonal transport, and oxidative damage to the motor neuron. When mitochondrial ATP production is impaired, from mitochondrial dysfunction compounded by stressors of oxidative damage, the motor neuron will also have deficits in axonal transport of proteins, organelles, and neurotransmitters, resulting in progressive denervation of muscle fibers [108]. Fast progressors typically have high levels of mitochondrial DNA damage and the presence of lipid peroxidation markers (e.g., 4-HNE), suggesting metabolic stress is associated with rapid loss of MUNE. In the clinical setting, MUNE is being used as an endpoint in clinical trials evaluating ASO therapies, targeting SOD1 and C9orf72 variants, whereby it tracks the preservation of motor units by evaluating muscle recruitment in the event of treatment [109].

#### 3.3.2. Compound Muscle Action Potential: Tracking Functional Motor Unit Output and Neuromuscular Transmission

The CMAP amplitude is a measure of the electrical response of a muscle when it is electrically stimulated by a depolarization of the motor nerve, indicative of the functional motor unit output induced by the recruitment of motor units. For fast progressors, the CMAP amplitude declines rapidly over time due to the loss of motor units from motor unit dropout, synaptic dysfunction, and impaired neuromuscular transmission. Research has demonstrated that fast progressors exhibit reductions in CMAP of ≥30% during the first six months after symptom onset, an event that has also been associated with early evidence of respiratory decline and the need for ventilatory assistance. In other words, CMAP serves as a marker because it reflects the remaining physiological capacity of the motor units, while MUNE reflects the loss of individual motor units. This immediately makes CMAP a complementary biomarker for monitoring the clinical course of disease on progression [110].

While at a mechanistic level, CMAP reductions observed in fast progressors derive from the synaptic failure of motoneurons with the neuromuscular junction (NMJ)—a progression complicated further by glutamate excitotoxicity, mitochondrial dysfunction, and impaired synaptic vesicle release. It is known that excessive glutamate stimulation leads to calcium overload of motor neurons that cause mitochondrial damage and ultimately synaptic degeneration [111]. In ALS, the pathway for synaptic transmission is further diminished by impaired mitochondrial energy production in the case of SOD1 and FUS variants, heightening the CMAP decline in fast progressors. Recently, operationalizing the use of CMAP amplitude in parallel with NfL has been shown to improve the prediction of time to the need for NIV, providing further context to the integration of CMAP into multimodal biomarker approaches [112]. CMAP has been designated a secondary outcome in trials of neuroprotective and anti-inflammatory treatments where, in the context of the changes observed in CMAP amplitude, reduction of loss motor units and improvement in synaptic function are evident. For example, with the use of IL-6 inhibitors, stabilization of CMAP has been coupled with a reduction in neuroinflammation, further evidence of the role of CMAP as a marker of the effectiveness of treatment [113].

#### 3.3.3. High-Density Surface Electromyography: Mapping Motor Unit Firing Patterns and Neuromuscular Instability

HD-sEMG is a sophisticated electrophysiological device that generates spatially rich maps of motor unit recruitment and firing patterns and patterns of neuromuscular coordination. In contrast to traditional EMG, which only captures electrical signals from a select number of muscle fibers, HD-sEMG can capture widespread motor unit activity over large muscle regions, offering a more complete representation of neuromuscular integrity [114]. In fast progressors, HD-sEMG indicates abnormal motor unit recruitment, asynchronous firing patterns, and increased neuromuscular jitter, and all of those abnormalities are associated with rapid functional decline [115]. These electrical and mechanical abnormalities have their basis in disrupted synaptic transmission and the inability to regulate calcium at the NMJ. The excess of excitatory input caused by glutamate-mediated hyperexcitability disruption consists of hyperexcitable input and disrupts the implied coordination of firing action potentials in motor units, resulting in ineffective muscle contractions and further weakness [116]. Studies with HD-sEMG indicate fast progressors experience a pronounced desynchronization of motor units even within three months of diagnosis, giving HD-sEMG utility as a marker of early detection. Abnormalities detected by HD-sEMG are also associated with faster decline in limb and bulbar function, making them prognostically relevant [117].

HD-sEMG is also being used to assess whether this outcome is an applicable biomarker to monitor the efficacy of synaptic-targeting therapy, such as glutamate receptor antagonists and mitochondrial stabilizers. Interestingly, in clinical trials with riluzole and edaravone, HD-sEMG has demonstrated improvement in motor unit synchronization with slower ALSFRS-R decline [70].

#### 3.3.4. Single-Fiber Electromyography (SF-EMG) and Neuromuscular Jitter: Early Markers of Synaptic Failure

SF-EMG is a specialized technique used to identify neuromuscle jitter and transmission delay that occur at the NMJ. SF-EMG is most sensitive to early synaptic failure and is a major biomarker in fast progressors because they have a global NMJ pathology well before their overt muscle weakness is clinically apparent. Evidence of increased neuromuscular jitter was observed in fast progressors who are carriers of a C9orf72 and a TARDBP mutation, and this jitter is thought to be reflective of poor synaptic vesicle release and abnormal calcium flux [118].

At the molecular level, developments in neuromuscular jitter are a direct result of RNA-binding protein dysfunction, mitochondrial deficiency, and oxidative damage to their synaptic terminals. In patients with pathologically defined TDP-43, RNA-processing dysfunction is expected to interfere with the production of constitutive synaptic proteins, leading to syntheses being inefficient and poor neurotransmitter relief. Importantly, clinically relevant and elevated jitter identified through SF-EMG can assist in predicting early bulbar signs and rapid respiratory decline and thus help to establish SF-EMG as a specific biomarker for a plethora of clinical trials in patient stratification [82].

#### 3.3.5. Peripheral Nerve Excitability Tests: Monitoring Axonal Membrane Function and Ion Channel Activity

Peripheral nerve excitability tests measure axonal membrane engagement and ionic channel dysfunction and efficiency of neuromuscular transmission. There are normal variations of sodium, potassium, and calcium channel activity that are disrupted in ALS. In fast progressors with ALS, there is increased axonal excitability associated with promoted potassium channel dysfunction, which follows a pattern of increased axonal excitability, abnormal firing threshold, and decreased neuromuscular output. The decrement of abnormal nerve excitability profiles in fast progressors correlates with rapid declines in CMAP amplitudes and muscle strength [119].

From a therapeutic standpoint, peripheral nerve excitability tests are used to monitor the effects of ion-channel-targeted therapies: sodium channel blocker and potassium channel modulators. Therapies aim to restore normal axonal excitability, preventing motor unit recruitment and subsequently delaying functional decline [120].

#### 3.3.6. Emerging Techniques: Microelectrode Arrays and Neuromuscular Ultrasound

Microelectrode arrays (MEAs) and neuromuscular ultrasound are emerging as a pair of complementary tools to optimize precision in interpreting and relating electrophysiological properties. MEAs allow high-resolution recordings of single motor unit activity and can continuously assess multimotor unit recruitment and timing of motor unit firing [121]. Neuromuscular ultrasound allows visualization of muscle architecture and can detect motor unit atrophy, fibrotic muscle, and increased hyperechoic signals. Fast progressors demonstrate consistently observable muscle atrophy and structural abnormalities related to ultrasound findings that correlate with MUNE and CMAP declines [122].

#### 3.3.7. AI-Powered Electrophysiological Analysis and Multimodal Integration

AI-powered platforms are changing the way electrophysiological data are interpreted and can help assess large-scale MUNE, CMAP, and HD-sEMG datasets to facilitate the identification of motor unit dysfunction patterns hidden within the complexity of the underlying pathology. Current machine learning models are applied to predict time to ventilatory support and can help stratify patients based on their speed of progression. Importantly, the relationship of electrophysiological analysis and use of neuroimaging (e.g., DTI, MRI) and molecular biomarkers (e.g., NfL) is standardizing the recent advances of precision medicine in ALS and potentially real-time therapeutic ratification [123].

Electrophysiological biomarkers encompass a unique capacity to make real-time observations of motor neuron degeneration and neuromuscular dysfunction during fast progression of ALS. MUNE, CMAP, HD-sEMG, SF-EMG, and peripheral nerve excitability tests can provide a framework to characterize the complexity of synaptic failure, axonal degeneration, and motor unit loss, which are valuable for prognostic and early admission, as well as therapeutic monitoring [124]. Going forward, the progression of analysis bags using AI and use of multiplex biomarkers can integrate electrophysiological biomarkers into patients’ personalized experiences in care, as well as biomarker-driven clinical trials [125].

## 4. Genetic and Molecular Factors Driving Fast ALS Progression

Genetic variants and molecular dysregulation play a significant role within the heterogeneity of ALS. The factors that contribute to fast progression in fast progressors indicate a rapid functional decline or shortened survival within the ALS population. Fast progressors compared to typical ALS patients demonstrate the presence of highly penetrant genetic variants that produce neurotoxic cascades early on which leads to increased motor neuron death, mitochondrial dysfunction, and a reduced capacity to maintain protein homeostasis [126]. This category of fast ALS progressor has genetic variants known to be associated with fast progression phenotypes and distinguishable cellular mechanisms involved in them, such as SOD, C9orf72, FUS, and TARDBP. For instance, there are many variants in SOD1 which could lead to increased protein aggregation, variants in C9orf72 which could show deficits in autophagy, and variants in FUS which relate to glutamate excitotoxicity [127]. Each of these neurotoxic cascades or cellular mechanisms could provide an explanation for the rapid clinical process and clinical functional decline associated with the clinical course of fast ALS progression. The combination of genetic predisposition and environmental exposures (e.g., exposure to heavy metals and/or environmental toxins) could lead to the proposed phenotype or theoretically shape the disease phenotype, accelerating degeneration of the motor system. Recent literature has showcased the use of multiomic profiling studies and CRISPR-based functional genomic studies, thus expanding the genetic and molecular landscape of ALS while providing a new and timely roadmap for targeting therapeutic interventions for “fast” ALS progression [128].

### 4.1. SOD1 Variants: Oxidative Stress, Protein Misfolding, and Mitochondrial Dysfunction in Fast ALS Progressors

The genetic variants of the SOD1 gene are some of the most researched genetic causes of ALS, especially in regard to the manifestation of fast progressors, early-onset clinical signs, and the short period of survival. Among the >200 variants recorded, the A4V variant is one of the most accelerated forms of SOD1-associated disease, which produces periods of survival of less than 12 months [129,130]. Wild-type SOD1 is an antioxidant enzyme that protects neurons from ROS damage, however, mutant SOD1 is involved in toxic gain-of-function-related degeneration of motor neurons including defective protein misfolding, mitochondrial and ER dysfunction, ROS damage, and failure of autophagy. Newer studies also show all the harmful impacts of loss of function related to RBP, mislocalization of proteins, and gene–environment interactions on the progression of motor neuron degeneration in this neurodegenerative motor neuron disease, specifically with SOD1 variants and the focus on the complexity and multipathway nature of fast progression in this group [51].

#### 4.1.1. Toxic Gain-of-Function Mechanisms in SOD1 Variants: Protein Misfolding and Aggregation

SOD1 variants have some unique gain-of-function toxic mechanisms, namely, protein misfolding followed by protein aggregation. A key aspect of SOD1 variants is the gain-of-function toxicity that arises due to acquiring both misfolded/intact and aggregated SOD1 protein intracellularly. Misfolded SOD1 proteins will accumulate intracellularly through the formation of toxic and insoluble aggregates that result in damage to cell homeostasis. Misfolded SOD1 proteins inhibit normal chaperoned-mediated folding (HSP70, HSP90) and proteasomal degradation. In a healthy cell, molecular chaperones are able to correctly fold many proteins and inhibit aggregation [131]. The total amount of misfolded SOD1 will accumulate in cells and eventually overwhelm all the chaperone molecules and insoluble aggregates, leading to a swelling of the cells that will need to be cleared from the cells. Misfolded SOD1 aggregates also acts as a potent antagonist of 26S proteasomal activity by blocking damaged protein degradation while advancing toxic cellular debris from the damaged cell that is in a state of genomic crisis [132].

In addition to proteasomal function, misfolded SOD1 protein aggregates induce secondary interactions with other ALS proteins (i.e., TDP-43 and FUS) to support toxicity. Among various new studies, recent studies have shown that a misfolded SOD1 protein could actually cross-seed aggregation of TDP-43 and lead to cytoplasmic inclusions that impair RNA metabolism and even axonal transport [24]. FUS aggregates arise in an aggregated manner as associations with misfolded SOD1 which importantly confers a feed-forward loop of proteinopathy that thwarts fast progressors’ motor neuron degeneration fidelity. The coaggregation capabilities articulate the intermixed nature of protein-misfolding diseases and the realistic possibility that therapies targeting multifactorial pathways of protein aggregation may be necessary in the future [73].

Mitochondrial dysfunction is an improvement for SOD1-related ALS because it enhances rapidly degenerating motor neurons. In fact, mutant SOD1 aggregates in the mitochondrial intermembrane reduce the basic processes to create energy and coexist [133]. To create energy, the mutant SOD1 knocks out the membrane proteins in mitochondria necessary to thwart electron transport chain activity of complexes I and IV in the mitochondrial membrane, causing collectively decreased ATP activity and means of energy cycles for the motor neuron to have an acute energy crisis with the reduction of energy. The reduction of energy has disastrous implications when the motor neurons have the inability to award energy equal to the unit standard for vital processes with respect to axonal transport and synaptic transmission that inevitably results in a tragedy specifically with respect to a cell’s vulnerability in degeneration or progressive neuropathological diseases such as ALS [108].

Similar to neurons with Ca^2+^ overload associated with mutant SOD1, acute overload of energy leads to a good deal of cell death. Depressed Ca^2+^ homeostasis or mPTP activation retains a good deal of intracellular structure and targets the cytosol largely to derive cytochrome c disintegration to initiate apoptosis. Mitochondrial-associated membranes (MAMs) will disrupt abnormal Ca^2+^ misfolding of and lipid moving from mitochondria to the endoplasmic reticulum (ER) [134]. The disruption of MAMs has the potential to prolong Ca^2+^ dysregulation and importantly worsen the high cost of ER stress. The SOD1 aggregates manifest loss of MAM integrity, induce oxidative stress, and increase lipid peroxidation. There are a number of ways to ameliorate mitochondrial dysfunction, for example, SMs (i.e., SS-31) to stabilize membranes and pump out ATP to ameliorate disease progression in clinical models [135].

#### 4.1.2. Oxidative Stress and Reactive Oxygen Species Accumulation

The generation of ROS along with oxidative stress is one of the first and arguably the most recognized consequence of a SOD1 mutation. Mutant SOD1 loses the ability to effectively neutralize ROS and can also overproduce free radicals as discussed in the context of its unusual oxidative cycling step, such that it reverts back to a highly reactive state [136]. The overgeneration of superoxide, hydrogen peroxide, and hydroxyl radicals can result in oxidative damage to macromolecules in cells such as lipids, proteins, and DNA. Fast progressors routinely demonstrate increased markers of oxidative stress (higher levels of 8-OHdG as a marker of DNA damage and 4-HNE as a marker of lipid peroxidation) [137].

The oxidative-stress-induced damage has a direct impact on cytoskeletal stability and axonal degeneration. For example, oxidized neurofilaments disrupt normal axonal transport and ion localization and result in synaptic dysfunction that can ultimately result in motor neuron dropout [138]. The relationship that oxidative stress and mitochondrial function share is not trivial and occurs in a cycle, where the generation of ROS contributes to mitochondrial damage, then compromises ATP production capacity and ultimately disrupts cellular energy homeostasis [139]. Edaravone and other clinical trial and experimental agents that address mitochondrial ROS generation come from initial studies looking for an antioxidant-based regime to intervene in and ultimately disrupt this cycle to slow disease progression [140].

#### 4.1.3. RNA-Binding Protein Mislocalization and Splicing Errors

In addition to the previously mentioned effects on protein folding and oxidative stress, mutant SOD1 may lead to RNA-binding protein dysfunction related to TDP-43 and FUS. The misfolded aggregates of SOD1 induce the mislocalization of RBP away from the nucleus and into the cytoplasm, where they form toxic aggregates [69]. The mislocalization of RBP disrupts the splicing and transport of critical elements, including neurofilaments and synaptic proteins. A previous study shows that axonal transport defects due to disrupted RNA metabolism can lead to early motor dysfunction in fast progressors, who experience a faster decline on the ALSFRS-R scale and an earlier respiratory involvement. One potential therapeutic direction is restoring RNA homeostasis through various pathways, including small molecules that would modulate TDP-43 localization in SOD1-ALS patients [141].

#### 4.1.4. Gene–Environment Interactions: Amplifying Motor Neuron Vulnerability

Gene–environment interactions modulate the clinical expression of SOD1-ALS. This neurodegeneration-associated gene–environment interaction is primarily from the presence of environmental toxins and harmful oxidative stressors that amplify underlying genetic susceptibility [142]. Exposure to heavy metals like lead and forms of mercury modifies intracellular antioxidant defenses, like glutathione, increasing susceptibility to SOD1-induced oxidative injury within motor neurons. In addition to heavy metal exposure, pesticides and agricultural chemical exposures can induce lipid peroxidation and mitochondrial dysfunction while cigarette smoking can amplify systemic ROS by increasing oxidative stress [143]. Studies suggest that individuals with high environmental exposure to neurotoxic materials have a more rapid functional decline and reduced survival, indicating that reducing environmental exposures may help mitigate clinical severity [144].

#### 4.1.5. Liquid Biopsies and Circulating SOD1 Aggregates for Therapeutic Monitoring

Liquid biopsy technology advances have proposed the minimally invasive detection of SOD1-related ALS progression via liquid biopsy of misfolded SOD1 aggregates. EVs containing misfolded SOD1 and RNA cargo have been found in the plasma and CSF of fast progressors and correlate with disease severity and therapeutic response. Increased circulating SOD1 aggregates are being evaluated as biomarkers for patient stratification for clinical trial participation using ASOs (e.g., tofersen) [145]. The complexity of SOD1-ALS likely requires a combination therapy approach that is multi-, if not completely, complementary for engendering a similar fate for the disease. ASO therapies like tofersen have been displayed to convincingly reduce SOD1 expression at the mRNA level with slow motor decline and decreases in NfL levels [146]. Mitochondrial-targeting compounds, drugs that increase autophagy, and antioxidants are being trialed in combination therapy to ultimately treat SOD1-ALS, consistently addressing protein aggregation, energy deficits, and oxidative stress together. CRISPR-based technologies designed to offer precise gene editing to permanently correct SOD1 variants, or silence SOD1 variants using base editing, and CRISPR interference (CRISPRi) strategies offer promising future developments to permanently correct or silence disease-associated SOD1 variants [147]. The potential mechanisms involved in SOD1-ALS have distinctly highlighted the convergence of events and stressors such as protein misfolding, mitochondrial dysfunction, oxidative damage, and RNA-binding protein dysfunction are all found in fast disease progression. Multiomics approaches, biomarker-identified associations, and targeted therapies suggests future research may personalize treatment regimens, while also developing curative approaches for SOD1 mutation carriers. These novel developments will lead to smart biomarker-driven therapeutic approaches and preventative measures aimed primarily at decreasing environmental toxicity exposure to improve patient outcomes over time [148].

### 4.2. C9orf72 Expansions: Toxic RNA, Dipeptide Repeat Proteins, and Nucleocytoplasmic Transport Dysfunction

The C9orf72 HRE is the most common genetic cause of ALS and frontotemporal dementia (FTD) and is uniquely associated with rapid clinical disease progression, onset of symptoms at a young age, and a shorter life span. In a healthy person, GGGGCC (G4C2) repeats typically range from 2 to 30 copies; however, in affected individuals, HREs expand from hundreds to thousands of copies, initiating a multistep cascade of molecular events that dysregulate many cellular biological processes [149]. Pathological processes include RNA foci formation, dipeptide repeat protein toxicity, nucleocytoplasmic transport disruption, and autophagy defects. Collectively, the toxic potential of these processes leads to more rapid degeneration of motor neurons through dysregulated RNA metabolism, proteostasis, mitochondrial function, and synaptic integrity. Furthermore, interesting findings related to R-loop formation, chromatin remodeling, and immune deregulation add complexity to our understanding of the degenerative mechanisms of C9orf72, making C9orf72-ALS one of the most mechanistically diverse and therapeutically challenging forms of ALS [150].

#### 4.2.1. RNA Toxicity: Formation of RNA Foci and Sequestration of RNA-Binding Proteins

A defining and classic pathology of C9orf72 pathological mechanisms of ALS is the formation of RNA foci in the nuclei of motor neurons and glial cells, caused by bidirectional transcription of expanded G4C2 repeats. Toxic RNA foci act to sequester many critical RNA-binding proteins (RBPs) from functional pools, such as TDP-43, FUS, hnRNP H, and ADARB2 that are responsible for RNA splicing, transport, and stability [151]. The sequestering of these important RBPs leads to widespread disruption of alternative splicing and myriad misspliced transcripts that interfere with processes required for synaptic transmission, mitochondrial, and proteostasis. Research utilizing motor neuron models derived from patients demonstrated that the misregulation of important splicing targets (e.g., axonal transport, neurofilament assembly) can also contribute to accelerated axonal degeneration and neuromuscular dysfunction [152,153].

#### 4.2.2. Dipeptide Repeat Proteins: Aggregation, Stress Granule Disruption, and Cellular Toxicity

DPRs are made via repeat-associated non-AUG (RAN) translation from the expanded G4C2 repeats and there are five species of DPR (poly-GA, poly-GR, poly-GP, poly-PR, and poly-PA). The most toxic species, poly-GA, poly-GR, poly-PR, are linked to cellular toxicity by aggregation, defects in proteostasis, nucleolar stress, and mitochondrial dysfunction culminating in neurodegeneration [154]. The overwhelming overall effects of poly-GA are related to aggregation that have been shown to form large fibrillar inclusions in the cytoplasm leading to inhibition of motor neuron function by their sequestration of components of the ubiquitin–proteasome system (UPS) and autophagy machinery, ultimately shown to result in proteostasis and subsequent increases in misfolded proteins. Further research showed that the accumulation of poly-GA aggregates was correlated with early motor function declines measured by ALSFRS-R and rates of decline in fast progressors [155].

In contrast, poly-GR and poly-PR localize to the cytoplasm and nucleus and impair nuclear function and ribosomal biogenesis. The arginine rich DPRs, poly-GR and poly-PR are also shown to disrupt ribosome assembly in the nucleolus in response to cellular stress, by interacting with nucleolar proteins, such as NPM1 and fibrillarin [156]. With cryo-electron microscopy (cryo-EM), poly-PR and poly-GR were also observed to bind directly with nucleoporins inside of the nuclear pore complex (NPC), which obstructs nucleocytoplasmic transport and causes increased toxicity. Additionally, poly-PR also blocked initiation of stress granules, preventing motor neurons from making an effective neuroprotective response to cellular stress. Possible therapeutic interventions that target toxicities of DPRs could be multiple DPR-specific antibodies and/or small molecules that inhibit RAN translation [157].

#### 4.2.3. Nucleocytoplasmic Transport Dysfunction: Disruption of Nuclear Pore Complex (NPC) Integrity

Disruption of nucleocytoplasmic transport from toxic RNA foci and DPRs play an important pathological role in C9orf72-related pathology and ALS. In normal cellular health, NPCs regulate and maintain the bidirectionality and trafficking of proteins, RNA, and other macromolecules into and out of the nucleus to maintain homeostasis within the cell. Yet, poly-GR and poly-PR in fast progressors have altered nucleoplasmic-transport-related cycles leading to structural deformities of the NPCs (e.g., NUP62, NUP98, NUP205) and aggravate their inability to carry out their functional purpose. This NPC malfunction and the presence of toxic proteins and RNAs allow important mislocalized nuclear proteins coordinating genomic integrity, DNA repair, and stress response and preventing toxic proteins and RNAs from entering the nucleus. Thus, mislocalized nuclear proteins, toxic proteins, and RNAs occupy the nucleus. This structural dysfunction indicates serious nuclear stress, genome instability, and neuronal apoptosis that cause rapid loss of MNs [158].

Recent studies using super-resolution imaging indicate that distinct nucleoporopathy pathology occurs early in disease progression and may represent suitable biomarkers for early diagnosis in fast progressors. Pharmacological treatment strategies aiming to restore nucleocytoplasmic transport are based on small molecules to stabilize the structure of the NPC and chaperone-based therapies to prevent improper localization of nucleoporins or restore protein trafficking.

#### 4.2.4. Loss of C9orf72 Function: Autophagy Failure and Immune Dysregulation

While the majority of pathogenic mechanisms associated with C9orf72-related ALS are through toxic gain-of-function mechanisms, a loss of normal C9orf72 function, which has a detrimental effect on normal autophagic and lysosomal degradation, is also integral to disease progression. The C9orf72 protein competes with SMCR8 and WDR41 to regulate both autophagosome maturation and lysosomal trafficking [159]. In patient samples with fast progression, the consequence of loss of C9orf72 is that it completely inhibits successful maturation of autophagic vesicles into lysosomes, resulting in repressed organelle and protein aggregate clearance, being unrecoverable from misfolded proteins, and loss of autophagic vesicles before successful clearance of these faulty autophagic vesicles, which is detrimental to stress and vulnerability of that motor neuron cell type.

Pathologically, the loss of function of C9orf72 could only be compounded by the loss of function of C9orf72, contributing to chronic neuroinflammation by dysregulated microglial activation, where C9orf72-deficient microglia secrete increased quantities of cytokines, such as IL-1β and TNF-α, that generate a toxic microenvironment over time, with the tendency toward neuronal toxicity. Therapeutic strategies to enhance autophagy and neuroprotective effects, such as autophagy-inducing drugs (e.g., rapamycin, trehalose) and lysosomal stabilizers examined to date, will likely slow the disease trajectory, restore proteostasis to baseline, and mitigate neuroinflammation [160].

#### 4.2.5. Gene–Environment Interactions: Accelerating Neurodegeneration in C9orf72 Expansion Carriers

Environmental factors, such as pesticide exposure, heavy metals, air pollution, and smoking, have demonstrated a hyperexacerbation of C9orf72-related ALS through reactive oxidative stress, DNA damage, and breakdown of proteostasis [161]. Pesticides and herbicides work by a host of compounds to accelerate lipid peroxidation and mitochondrial dysfunction. Lead and mercury reduce glutathione, impairing the cell’s ability to scavenge ROS. Furthermore, smoking promotes DPR toxicity and nucleolar stress through increased systemic ROS generation [162]. Thus, gene–environment interactions are crucial to understanding the faster disorder progress in some C9orf72 carriers and whether environmental mitigation strategies could complement genetic therapies [163].

#### 4.2.6. Liquid Biopsies and Circulating Biomarkers: Non-Invasive Monitoring of Disease Progression

Liquid biopsy technologies present a meaningful way of monitoring C9orf72-related ALS and identifying circulating biomarkers. For example, elevated levels of DPRs, phosphorylated TDP-43, and toxic RNA species have been identified in the plasma and CSF of fast progressors. Additionally, EVs that carry poly-GA, poly-PR, and RNA that maintain that association with faster ALSFRS-R declines and mortality can also assist in stratifying patients for a clinical trial or quantifying therapeutic response [164].

#### 4.2.7. Therapeutic Approaches: Targeting Multiple Mechanisms for Maximum Efficacy

C9orf72-related ALS can be targeted through therapeutic approaches that specifically address multiple interconnected mechanisms simultaneously:Target sense and antisense G4C2 transcription with ASOs to reduce RNA foci and toxic DPRs following transcription [165].DPR-targeting antibodies or small molecules to inhibit RAN translation or neutralize toxic DPRs [166].Chaperone-based therapy or small molecules to facilitate restoration of nucleocytoplasmic transport associated with stabilization of the NPC [167].Autophagy inducers and lysosomal stabilizers to enhance clearance of cytotoxic protein aggregates and damaged organelles [168].

C9orf72 expansions result in rapid ALS through a multifaceted combination of RNA toxicity, DPR aggregation, nucleocytoplasmic transport dysregulation, and failure of autophagy. What is emerging in parallel is the identification of novel biomarkers, along with multiomics methodologies and targeted therapeutics, which will streamline and personalize treatment strategies that target both gain-of-function and loss-of-function mechanisms to improve clinical outcomes and perhaps slow disease progression [72].

### 4.3. FUS Variants: Early-Onset ALS, RNA Dysregulation, and Proteinopathy in Fast Progressors

FUS variants are an established cause of juvenile or early-onset rapidly progressive ALS. In this manuscript we will follow HGVS and designate variants at their first instances as NM004960.4:c.1574C>T p.(Pro525Leu) and NM004960.4:c.1562G>A p.(Arg521His); thereafter, in the interest of readability, we shall employ protein short forms (P525L, R521H). These genotypes are associated with rapid functional collapse and short survival—specifically, carriers of p.(Pro525Leu) have a median survival of ~12–18 months [169]. FUS is an RNA-binding protein involved modestly in RNA splicing and RNA transport and more substantially in DNA damage repair, stress granule dynamics, and regulation of autophagy. FUS-ALS pathology can be characterized by: (i) cytoplasmic misplacement of the FUS protein; (ii) cytoplasmic aggregation; (iii) RNA-processing perturbation; (iv) failed DNA repair; and (v) abnormalities in stress granule assembly/disassembly that may contribute to motor neuron degeneration. Collectively, it appears that converging mechanisms—including posttranslational modification cascades, cross-talk with other ALS protein systems including TDP-43, and impaired autophagy—can exacerbate toxicity and thereby provide putative testable therapeutic targets [170].

#### 4.3.1. Cytoplasmic Mislocalization, Aggregation, and Toxic Gain-of-Function Mechanisms

One feature that uniquely describes leukofuscinosis-associated ALS is cytoplasmic mislocalization and aggregation of FUS itself (the FUS variants affect the altered nuclear localization signal (NLS) of FUS). When localized in the nucleus FUS performs mRNA splicing and transcription and participates in responses to DNA damage processes, to name a few, primarily via usage of the FUS NLS. The cerebellar complex phenotype factors that are present at the beginning since disease onset and other neurological features of FUS variants, such as P525L and R521H, may be a result of loss of nuclear import mediated by the variants, leading to accumulation of FUS in the cytoplasm and the formation of insoluble aggregates that may be toxic to cells [171]. A rough assessment of the degree of cytoplasmic FUS mislocalization can be gauged via a continuum of severity estimated by the aggressive phenotypes associated with P525L variants. Upon mislocalization, FUS undergoes self-aggregation and amyloid-like fibril formation and can also form amyloid-like fibrils when interacting with other proteins in complex mechanisms with stress granules and ribonucleoprotein particles [172]. Studies using cryo-EM methods have revealed aggregates of FUS to take on a cross-β-sheet configuration, which is characteristic of amyloid fibrils, and demonstrate that FUS has the requisite features to promote aggregation and enmeshing of other misfolded proteins [173]. The aggregates lead to widespread disruption of the RNA metabolism, by sequestering RNA binding proteins, untranslated mRNAs, and components of stress granules, leading to an aggregation and neurotoxicity feed-forward loop [174]. New therapies are emerging for a variety of reasons, including small molecules that target β-sheet formation or therapeutic exercise with molecular chaperones to promote disaggregation, and are in preclinical evidence of efficacy studies [175].

#### 4.3.2. Posttranslational Modifications: Regulators of FUS Aggregation and Toxicity

PTMs are crucial in regulating FUS’s function, as well as its mislocalization and aggregation. Aberrant PTMs, alteration of phosphorylation, acetylation, and arginine methylation have been implicated in the pathogenesis of FUS-mediated ALS. Phosphorylation of serine residues on FUS, and in the case of the prion-like low-complexity domain in particular, demonstrates loss of RNA-binding ability and increases aggregation of cytoplasmic FUS. Hyperphosphorylated FUS was detected in ALS subjects’ post mortem spinal cord tissue and correlated with disease progression [176]. Arginine methylation, and again in particular within the low-complexity domain, has been shown to regulate FUS’s RNA-binding properties to sequester within stress granules. The hypocitrullination/hypercitrullination of key arginine residues allows for pathological aggregation that alters FUS’s normal phase-separation behavior that is essential for both phase separation of functional stress granules and disaggregation of pathological FUS [177]. Acetylation of lysine residues also may mediate loss of nuclear localization ability and encourage cytoplasmic formation of fibrils. Interventions targeting PTMs (e.g., methyltransferase inhibitors, deacetylase modulators) could offer therapeutic benefit in disorders characterized by FUS aggregation by halting or ameliorating FUS aberrant aggregation while restoring its normal cellular functions [178].

#### 4.3.3. RNA Dysregulation and Missplicing Events in FUS-Related ALS

The cytoplasmic mislocalization of FUS manifests in the different nuclear functions of FUS being lost, consequently leading to diverse RNA dysregulation, splicing defects, and impairments of RNA transport. In normal cellular function, FUS performs a fundamental role in the processing of RNAs involved in axonal maintenance, synaptic transmission, and mitochondrial function. Missplicing from mRNA dysregulation due to FUS expressed in ALS represents an additional motor neuron vulnerability for ALS. RNA sequencing of motor neurons derived from FUS-mutant iPSCs is revealing extensive splicing issues in important genes involved in gene expression and axonal and mitochondrial integrity, such as KIF5A, STMN2, and MFN2 [179]. Missplicing KIF5A (a cargo motor transporting protein) impairs the axonal transport of mitochondria, synaptic vesicles, and general cargo, contributing to axonal degeneration and synaptic dysfunction. Missplicing STMN2 (an axonal regeneration protein) also appears to reduce axonal repair capacity and was associated with more rapid functional decline in ALS patients [180]. In many instances, mislocalization of FUS can also affect mitochondrial dynamics/modification through missplicing of MFN2 and OPA1 and ultimately lead to fragmentation of mitochondria, reduced ATP production, and increased oxidative stress. Therapeutically correcting splicing defects via ASOs or RNA-targeted therapies is currently being investigated as a valid strategy [181].

#### 4.3.4. Defective DNA Damage Response and Genome Instability

In addition to this role in the regulation of RNA metabolism, FUS is also an important factor in directing the DNA damage response (DDR) when there is double-strand break that requires repair. Under normal circumstances, FUS is not only recruited to the damaged DNA site but also helps facilitate the recruiting of repair proteins and the assembly of the repair complex. In contrast, however, mutant FUS has defective DNA-binding capacity, delayed recruitment to the site of damage, and impaired DDR activation once it reaches the damaged DNA site [182]. While neurons using mutant FUS deal with DNA damage, it also may be the key contributor in a cycle of genomic instability and oxidative stress and there have been both patient and animal studies on FUS-associated ALS where the accelerated neurodegeneration is due to DNA double-strand breaks, persistent increases of ROS, and mitochondrial dysfunction, all promoting motor neuron viability and less motor neuron loss, and this accelerated disease progression is associated with this deficit causing a dysfunctional DDR process. Targeting the DDR pathways that are defective, whether through DNA repair enhancers or ROS scavengers, is a possible therapeutic route to reduce DNA-damage-related genome instability and slow the progression of FUS-associated ALS [183].

#### 4.3.5. Stress Granule Dysregulation and Persistent Aggregation

Not only does FUS regulate the assembly and disassembly of stress granules, which are formed during cellular stress to protect untranslated mRNAs, it also decreases stress granule formation and aggregates in ALS patients where not only is mutant FUS aberrantly incorporated with stress granules that do not disassemble but also persistent aggregates resulting in toxic aggregates. Persistent stress granules also act like nucleation sites for FUS fibrils to aggregate past the point of no return and are a source of propagation of further FUS aggregation and cellular toxicity [184]. These granules sequester important RNA-binding proteins, untranslated mRNAs, and components of the translation machinery, impacting protein synthesis and RNA homeostasis. Defective stress granule disassembly can be further made more severe with defective autophagy and chaperones. Autophagic flux is reduced in FUS-mutant motor neurons, disrupting clearance to accumulate defective granules and sequestered misfolded proteins [185]. Strategies targeting stress granules include small molecules that induce disassembly of granules and enhance autophagic flux (e.g., rapamycin and trehalose) to enhance the clearance of aggregated FUS [186].

#### 4.3.6. Cross-Talk Between FUS, TDP-43, and C9orf72 Pathology

FUS aggregation is integrated into a web of molecular-scale cross-talk of ALS-linked protein systems whereby the aggregation and physiology of one protein system with the implicated other protein systems cause an evolution of effect such as TDP-43 aggregation with mislocalization of FUS resulting in toxic coaggregates that induce a response that would result in accumulation. Further DPR forms of poly-GR and poly-PR from the expansions in C9orf72 perturb this cross-talk and are known to inhibit transport inhibition between the nuclear and cytoplasmic compartments; therefore, suppressing the recovery of miscolocalized proteins from the cytoplasm and suppressing the tendency for an increased cytoplasmic accumulation of FUS [187]. The cross-talk in these individual systems and the loss of autophagy and accumulation of mislocalized proteins create a feed-forward loop of protein aggregation and RNA dysregulation associated with increasingly fast motor neuron death rates in fast progressors. Combination therapies that target both FUS-TDP-43 systems, such as FUS aggregation with TDP-43 aggregation and neutralization of DPR toxicity, are in development to block the toxic cross-talk effect by these multiple systems on human health, i.e., neurodegeneration [188].

#### 4.3.7. Emerging Biomarkers and Therapeutic Approaches for FUS-Related ALS

Emerging biomarkers specific to FUS-related ALS include misspliced transcripts (e.g., STMN2, KIF5A), FUS aggregates in CSF and plasma, and stress-granule-associated proteins. Elevated phosphorylated forms of FUS, and FUS-positive EVs, are correlated with greater functional decline and rates of disease progression and can help with early diagnosis and stratification of patients for clinical trials [184]. Therapeutically, there are small molecules in development to inhibit FUS aggregation by targeting β-sheet formation (and preventing formation of fibrils) and ASOs that correct missplicing events to return transcripts to normal processing. Autophagy inducers and chaperone-based therapies are also being developed to increase the clearance of FUS aggregates and defective stress granules. Because FUS is involved with DNA repair, DNA repair enhancers or ROS scavengers are being developed to help with DNA instability and oxidative species. With these different RNA-based therapies coupled with targeted aggregation, RNA- and chaperone-based therapies of cellular-stress-related processes can collectively lead to lowering rates of disease progression with clinical outcomes that could be warranted [132].

FUS variants have been shown to lead to early-onset and fast-progressing ALS through a combination of cytoplasmic mislocalization, toxic protein aggregation, RNA dysregulation, and defective DNA repair. Future research can build towards slowing disease progression and improving outcomes among patients with aggressive FUS-related ALS pathways with methods using novel biomarkers, PTM targeting, and with a combination of emerging RNA and autophagy enhancer therapies [189].

### 4.4. TARDBP Variants: TDP-43 Proteinopathy, RNA Dysregulation, and Neurodegeneration in Fast ALS Progressors

TARDBP gene variants, that encode the transactivation response DNA-binding protein, 43 kDa, represent a genetic component of ALS causation and are particularly linked to rapid course, early respiratory failure, and severe upper and lower motor neuron loss. TDP-43 is a critical RNA-binding protein that carries out various cellular functions, such as RNA splicing, transport, stability, translation, and the regulation of stress responses. In ALS, however, variants in the TARDBP gene or aberrant PTMs can cause TDP-43 to mislocalize from the nucleus to the cytoplasm, where TDP-43 forms inclusions that are toxic [190]. Mislocalization of the TDP-43 protein has two significant pathological effects: first, loss of normal nuclear functions and extensive RNA dysregulation; second, gaining toxic function due to aggregation in the cytoplasm. Functionally, this drives the underlying accelerated neurodegenerative process that is noted in rapid progressors, with new evidence showing a role for PTMs, coaggregation with other ALS-related proteins, and mitochondrial dysfunction all contributing to this atypical clinical phenotype [191].

#### 4.4.1. Proteins Shifting into the Cytoplasm, Clumping Together, and Acquiring Harmful New Functions

Perhaps the most definitive pathological feature of TARDBP-related ALS is the mislocalization of TDP-43 from the nucleus to the cytoplasm resulting in soluble and insoluble inclusions. Variants in TDP-43, which most commonly affect regions located in the C-terminus, likely influence TDP-43 nucleocytoplasmic localization signals (NLSs), hampering NLS-induced import for TDP-43 subcellular localization and potentially accumulating TDP-43 in the cytoplasm. Following cytoplasmic mislocalization, TDP-43 becomes subject to several PTMs, phosphorylated at many sites (particularly serines S409/S410) as well as undergoing ubiquitination and proteolytic cleavage into toxic C-terminal fragments (CTFs) [192]. These changes could enhance the aggregation into phosphorylated TDP-43 (pTDP-43)-rich inclusions that are resistant to degradation by the UPS and autophagy; in fact, cryo-EM studies show aggregates that contain the cross-β-sheet structure that is a hallmark of the amyloid fibrils and further stable aggregates have an increased propensity for aggregation [193].

The toxic gain-of-function mechanisms of cytoplasmic TDP-43 aggregation sequester RBP, untranslated RNAs, and important components of stress granules and negatively interfere with normal biology. The inclusion of an abnormally aggregated toxicity and TDP-43 aggregates impeded protein homeostasis, and TDP-43 induced mitochondrial dysfunction and oxidative stress created a neurotoxic cycle that perpetuated the process [170,184]. CTFs of TDP-43 (25 kDa, 35 kDa) had greater aggregative potential than full-length TDP-43, which functioned as a nucleation site for the formation of amyloid-like fibrils; furthermore, CTFs and pTDP-43 were correlated with faster decline in ALSFRS-R and an earlier onset of respiratory failure. CTFs and pTDP-43 are emerging important biomarkers for patient stratification, and therapeutic approaches are being developed that attempt to construct small molecules that could inhibit TDP-43 phosphorylation and aggregate and/or molecular chaperones that could extend the disaggregation or clearance of toxic inclusions [194].

#### 4.4.2. Loss of Nuclear TDP-43 Function: RNA Dysregulation and Splicing Defects

The nuclear loss of TDP-43 results in widespread RNA dysregulation, then perturbs the splicing, transport, and stability of thousands of RNA transcripts, many critical for motor neurons. TDP-43 binds UG-rich regions of pre-mRNA, broadly regulates alternative splicing, and specifies which mRNAs will have designated cryptic exons to be spliced out. The nuclear loss of TDP-43 causes missplicing of critical genes related to axon maintenance and development, synaptic function, mitochondrial homeostasis. One major consequence is aberrant splicing of STMN2, which encodes a primary protein of axonal regeneration and repair. In patients with unknown coding relationships, diminished expression of STMN2 with inclusion of the cryptic exon was associated with faster decline of motor function and poorer axonal regeneration in fast progressors [195].

In addition to STMN2, loss of TDP-43 identity is implicated in ongoing splicing with respect to stability of KIF5A (essential for axonal transport) and NEFH (neurofilament heavy chain) and mitochondrial genes (MFN2 and OPA1). Abnormal splicing of KIF5A reduces transport of synaptic vesicles and mitochondria, which magnifies degenerative processes toward axonal degeneration. TDP-43 loss of function was associated with build-up of long non-coding RNAs and cryptic exons with further transcriptionally dysregulated features. The development of RNA-targeted therapies (ASOs) created to target and correct aberration in splicing and restore typical RNA processing would provide rationale for a new treatment consideration towards ALS associated with TARDBP [180].

#### 4.4.3. Posttranslational Modifications and the Amplification of TDP-43 Toxicity

PTMs are significant determinants of TDP-43 stability, keeping in mind TDP-43 localization, assembly, and aggregations. TDP-43 aberrant PTMs (phosphorylation, ubiquitination, acetylation, and proteolytic cleavage) increase gain-of-function toxicity. It has been established that TDP-43 accessibility to phosphorylation at serine residue sites (S409/S410) will begin aggregation into an insoluble form, while ubiquitination occurs in UPS degradation mechanisms [196]. TDP-43 aggregates with hyperubiquitination may indicate that excessive non-essential aggregates overloaded proteasomal degradation such that the proteasomal system failed, allowing the toxic aggregates to persist. Acetylation at lysine residues in the RNA-binding domain further inhibits TDP-43’s binding to and regulation of target RNAs and also promotes mislocalization and aggregation. Proteolytic cleavage by caspases and calpains can produce CTFs with a higher aggregation propensity, as they can also be seeding centers for further fibril production [197].

Targeting PTMs has emerged as an exciting therapeutic target, as some small molecules that inhibit TDP-43 phosphorylation or promote clearance (e.g., proteasomal or autophagic degradation) are currently being investigated in preclinical and early-phase clinical trials. And there are molecular chaperones that can reverse TDP-43 aggregation or potentially mitigate toxicity associated with PTMs that may have potential benefit for fast progressors [198].

#### 4.4.4. Mitochondrial Dysfunction and Oxidative Stress

The TDP-43 pathology is tightly linked with mitochondrial dysfunction and oxidative stress that contributes to the rapid neurodegeneration seen in fast progressors. Mislocalized TDP-43 is associated with interactions that occur directly with mitochondrial outer membrane proteins, including TOM20 and prevents nucleus-encoded mitochondrial proteins from entering the mitochondria, blocking their movement along the TOM pathway, lipid import, and oxidative phosphorylation. This impairs ATP synthesis and disrupts calcium buffering while increasing ROS production [199]. TDP-43 aggregates in mitochondria may also contribute to dysregulated expression of mitochondrial fusion and fission proteins (e.g., MFN2 and DRP1) which results in mitochondrial fragmentation and impaired mitochondrial dynamics altogether. This further creates deficits for motor neurons as they become more at risk for excitotoxicity or programmed cell death from apoptosis. The development of therapeutic strategies targeting mitochondrial function include mitochondrial stabilizers (e.g., SS-31) and antioxidant-based drugs targeting ROS for their main function. The interconnected processes—protein aggregation and misfolding, mitochondrial dysfunction, impaired vesicle trafficking—underlie the common axonal transport defects and degeneration of motor neurons in ALS fast progressors [200]. The figure below (Figure 2) intends to illustrate how genetic variants disrupt cellular homeostasis and promote neurodegeneration through multiple converging mechanisms.

#### 4.4.5. Cross-Talk with Other ALS-Related Proteins (FUS and C9orf72 DPRs)

There is often a synergistic cross-talk between TDP-43 pathology and other ALS proteins; specifically FUS and the dipeptide repeat proteins associated with C9orf72 expansions. Mislocalized FUS can coaggregate with TDP-43, promoting toxic inclusion formation and even further disturbances in RNA metabolism. Similarly, poly-GR and poly-PR DPRs will disrupt nucleocytoplasmic transport and prevent TDP-43 from returning to the nucleus, thus enhancing TDP-43 retention in the cytoplasm and aggregation. This synergy emphasizes a forward-feedback loop of proteinopathy, RNA dysregulation, and cell stress, furthering and amplifying the rate of degeneration in a fast progressor. Developing combination therapies that target TDP-43 and the aggregates of the associated proteins has considerable promise in interrupting this pathogenic cycle [201].

#### 4.4.6. Emerging Biomarkers and Therapeutic Approaches

Emerging biomarkers of TDP-43 pathology include pTDP-43 in CSF and plasma, cryptic exon inclusion events (e.g., STMN2 missplicing), and exosomes containing TDP-43 fragments. Increased levels of pTDP-43 in CSF and plasma predict increased rates of decline on the ALSFRS-R, suggesting that its measurement as a biomarker of earlier diagnosis of ALS or for therapeutic monitoring purposes may be useful. Furthermore, RNA-based biomarkers, including the inclusion of cryptic exons in mRNA transcripts, also hold potential value for diagnostic purposes. Currently, there are numerous treatments being explored targeting TDP-43 aggregation, ASOs for correcting RNA missplicing, or using autophagy to promote clearance of TDP-43 aggregates. Combination therapy (multiple monotherapies targeting these pathologies) has particular promise for fast progressors [202].

TARDBP variants underpin fast progression in ALS in a diverse and complicated manner involving TDP-43 mislocalization, toxic aggregation, RNA dysregulation, mitochondrial dysfunction, and poor protein clearance. Future therapeutic strategies will address TARDBP-associated ALS pathways and underlying mechanisms of all disease mechanisms, particularly regarding cross-talk and PTMs, and the dysregulation in RNA processing. Targeted therapies hold the opportunity to slow or halt disease progression and lead to improved outcomes in this aggressive subtype of ALS [132].

### 4.5. ATXN2 Expansions: Intermediate PolyQ Expansions and Increased ALS Risk

Expansions of the Ataxin-2 (ATXN2) gene are a significant genetic risk modifier for ALS. Normal alleles have ≤22 CAGs, whereas intermediate expansions of ~27–33 CAGs increase the risk for ALS by ~5–7×, cause earlier onset, and increase the rate of progression. Expansions of ≥34 CAGs with normal CAGs cause the spinocerebellar ataxia type 2 (SCA2). There are mechanisms by which intermediate repeat variants converge toward TDP-43 proteinopathy, stress granule dysfunction, defects in RNA metabolism, impaired autophagy, and mitochondrial stress (see reference [203]). Additional functions for Ataxin-2 relate to cytoskeletal dynamics, calcium handling, and mitochondrial RNA metabolism. Many therapeutic avenues exist: ATXN2-targeted ASOs, autophagy enhancers, and inhibitors of aberrant phase separation [203,204].

#### 4.5.1. Intermediate PolyQ Expansions: A Genetic Modifier of ALS Risk

Genetics, in familial and sporadic disease, consistently connect ATXN2 27–33 CAGs with increased rates of ALS. The toxicity observed is length-dependent and can enhance risk in combination with variants in TARDBP or C9orf72. The dose-dependence of polyQ toxicity supports the targeting of Ataxin-2 directly: preclinical knockdowns of Ataxin-2 with ASOs, reduction in TDP-43 aggregation, and improvement in motor outcomes in animal models [204,205,206].

#### 4.5.2. Phase Separation and Aberrant Liquid-to-Solid Transitions in Stress Granules

Ataxin-2 plays a role in the assembly of stress granules via liquid–liquid phase separation. PolyQ expansions shift granules towards less fluid, more solid, states, that can trap RBPs and ultimately disrupt homeostasis of RNA; fluorescence recovery after photobleaching (FRAP) and cryogenic electron microscopy (cryo-EM) both show less fluidity and intermolecular amyloid-like fibrils in Ataxin-2-polyQ granules. From a therapeutic perspective, small-molecule modulators of phase behavior—ultimately inhibitors of pathological solidification or stabilizers of physiological dynamics—may provide a rationale to restore granule turnover [207,208,209].

#### 4.5.3. Cross-Talk Between ATXN2 Expansions and TDP-43 Proteinopathy

Expanded Ataxin-2 increases TDP-43 occupancy time in stress granules while promoting mislocalization to the cytoplasm, enabling aggregation and downstream RNA-splicing dysfunction. These transcriptomic studies have documented missplicing of STMN2, MFN2, and KIF5A that align with TDP-43 dysfunction. In various model systems, knockdown of Ataxin-2 decreased TDP-43 aggregates and normalized splicing, supporting the potential for a causal interaction that is now being explored clinically [210,211,212,213].

#### 4.5.4. Non-Canonical Roles of Ataxin-2: Cytoskeletal Regulation, Calcium Homeostasis, and Mitochondrial RNA Stability

Ataxin-2 interacts with actin/microtubule transport machinery that regulates axonal trafficking of mitochondria and vesicles; polyQ expansion also impacts flow and synaptic integrity. Additionally, it affects ER–mitochondria calcium transfer at MAMs, driving mitochondrial swelling, excess ROS, and apoptosis, and it binds mitochondrial mRNAs where expansion inhibits respiratory-chain translation and bioenergetics. These axes justify trials of cytoskeletal stabilizers, calcium modulators, and protectors of mitochondrial RNA biology [203,214,215].

#### 4.5.5. Autophagy Failure and Impaired Proteostasis

PolyQ-expanded Ataxin-2 also affects p62/SQSTM1 and LC3, disrupting autophagosome–lysosome fusion (and thus autophagic flux), leading to the accumulation of autophagic vesicles, TDP-43–positive inclusions, oxidative stress, and aberrant mitophagy with secondary ATP depletion and increased ROS. Autophagy and mitophagy inducers (e.g., rapamycin, trehalose) are being tested to try to restore clearance of aggregates and cellular homeostasis [216,217].

#### 4.5.6. Emerging Biomarkers and Therapeutic Strategies

Potential biomarkers include Ataxin-2 aggregates, misspliced transcripts, e.g., STMN2, KIF5A, and pTDP-43, in, e.g., EVs and CSF; EV cargo may track with severity and progression. Therapeutic approaches include ATXN2 ASOs, modulators of phase separation, autophagy inducers, and RNA targeting approaches, and combinations will be designed to contain both the toxicity of Ataxin-2 and TDP-43 pathology. As the interventions progress, EV-based and transcriptomic biomarkers would facilitate individualized dosing and monitoring in real time in ATXN2-expanded ALS [206,218,219].

## 5. Potential Therapeutic Approaches for Fast Progressors in ALS

Fast progressors in ALS are uniquely challenged by their rapidly accelerating disease course, early respiratory involvement, and lesser responsiveness to established therapies. Thus, fast progressors need therapies that target the underlying molecular drivers of rapid motor neuron loss, including neuroinflammation, mitochondrial dysfunction, oxidative stress, impaired RNA metabolism, and protein misfolding [220]. As insights into ALS pathophysiology develop, new and exciting therapies addressing the molecular and genetic drivers of ALS, including genetic correction, neuroprotection, and regenerative medicine, have begun to emerge and provide hope to fast progressors to decelerate their disease. A few other treatments will utilize multimodal and precision therapies, given the genetic and biomarker profiles of the patients, to usher in a new treatment pipeline for ALS, especially for genetic variants impacting SOD1, C9orf72, TARDBP, ATXN2, and FUS [221].

### 5.1. FDA-Approved and Off-Label Drug Treatments: Foundation of Neuroprotection

FDA-approved therapies are the backbone of ALS treatment targeting specific mechanisms, including glutamate excitotoxicity, oxidative stress, mitochondrial dysfunction, and ER stress. For fast progressors, the defect’s rapid accumulation of cellular damage, protein misfolding, and metabolic dysfunction may hinder the extent of therapeutic effect from standard therapies [111]. New combinations of combinatorial treatments, off-label use of drugs, and biomarker-driven treatment regimens are being developed to improve the efficiency of therapies at maximal efficacy, while limiting disease progression, by targeting multiple linked pathways that are often impacted together [222].

Riluzole, the first FDA-approved drug for ALS, is a glutamate-releasing antagonist that primarily inhibits the presynaptic release of glutamate and the conduction of voltage-gated sodium channels to inhibit excitotoxicity and hyperexcitability of neurons. While the contribution of riluzole to survival is small, having only been shown to improve survival by about 2–3 months on average, riluzole remains the backbone of ALS therapy, especially now that some combination therapies with complementary agents are emerging as potentially superior to a single agent [21,223,224]. There may also be a larger advantage to fast progressors who are generally subject to greater excitotoxicity when combination regimens involving riluzole and neuroprotective drugs, antioxidants, or mitochondrial stabilizers are considered. Clinical experiences have suggested that the use of riluzole together with sodium channel blockers (NaV1.6 inhibitors) plus glutamate clearance enhancers (e.g., ceftriaxone) or PB/TURSO may confer greater efficacy for preventing motor neuron death [225].

A new intravenous antioxidant, edaravone, was approved for use in Japan, the US, and Europe. Edaravone works by targeting ROS-induced damage by neutralizing free radicals and preventing oxidative stress directed upon the neuron. Edaravone has been shown to slow functional decline in early-stage ALS subjects, however, due to the burden of ROS, mitochondrial dysfunction, and protein aggregation, there will be little benefit for fast progressors [226]. This generation of evidence also raises the possibility that edaravone may be more effective when used in combination with mitochondrial-directed therapies (e.g., SS-31 (elamipretide), a mitochondrial-stabilizing peptide) because both elamipretide and edaravone guard against oxidative stress, work to preserve mitochondrial integrity, and also preserve ATP production [227]. Preclinically, SS-31 has been shown to improve mitochondrial membrane potential, decrease mitochondrial permeability, and block cytochrome c release, further protecting cells from apoptosis [228].

As mentioned previously, a breakthrough in terms of clinical treatments for ALS is sodium phenylbutyrate/taurursodiol (PB/TURSO; marketed as Relyvrio), which targets ER stress and mitochondrial dysfunction by stabilizing ER–mitochondria interactions. PB/TURSO acts to prevent the misregulation of calcium homeostasis, mitochondrial swelling, and corresponding engagement of the mPTP, which are the main processes by which apoptotic cell death occurs in fast progressors, in particular individuals with molecular aberrations in SOD1, C9orf72, and TARDBP [229]. Clinical trials have suggested PB/TURSO will provide a meaningful benefit to survival and retard progression, and hence it is a very promising treatment targeting fast ALS progressors most likely to have dysfunctional mitochondria. There are ongoing trials evaluating PB/TURSO with ASO drugs targeting the RNA toxicity and protein aggregation aspects of ALS for improved therapeutic effect [230].

Off-label uses of drugs may add potential benefit to approved drug therapies. Memantine (NMDA receptor antagonist), which can reduce synaptic excitability by preventing excessive glutamate-induced toxicity, and levetiracetam (a first-line drug for seizures) have been studied to reduce synaptic liabilities from unregulated excitotoxicity, which may be exaggerated by fast rates of progression from overactive glutamate signaling [231]. Memantine can act to reduce stimulation of NMDA receptors (but not by blocking the receptors) and excess calcium influx, and levetiracetam may act to inhibit presynaptic activities, associated with the synaptic vesicle, that lead to synaptic activity and excitotoxicity in motor neurons [232]. Most importantly, ceftriaxone may act to provide neuroprotection by functionally upregulating the GLT-1 transporter to increase clearance of glutamate, an important glutamate transporter, to reduce excitotoxicity [233].

#### 5.1.1. Emerging Combination Therapies and Synergistic Effects

For example, recent studies suggest that the combination of riluzole and NaV1.6 inhibitors may provide better neuroprotection by reducing injury from hyperexcitability in the synapse. In addition, sodium channel blockers could target the overactive NaV1.6 channels implicated in repetitive action potentials and excitotoxic neuronal injury associated with excess intracellular calcium in ALS motor neurons [234]. In the same line of thought, combinations of edaravone with mitochondrial protectors (SS-31 or mPTP inhibitors) have been shown to synergistically reduce ROS, preserve mitochondrial energy homeostasis, and inhibit cellular apoptotic capability [235].

#### 5.1.2. Biomarker-Guided Personalized Treatment Regimens

Biomarker-guided therapeutic monitoring is changing ALS management with the potential for real-time monitoring of disease progression and therapeutic responses, allowing for personalized treatment intervals for faster progressors. NfL, pTDP-43, and mitochondrial biomarkers provide guidance for axonal degeneration, TDP-43 protein aggregation, and mitochondrial dysfunction [236]. Increases in NfL, assessed from CSF and plasma, are associated with and correlate with axon degeneration and disability, thus making it an appropriate marker to measure the effectiveness of neuroprotective agents. For instance, declines in NfL in CSF or plasma after tofersen treatment show treatment effectiveness in SOD1-ALS [17].

Payloads of pTDP-43 can be treated as a biomarker for RNA-targeted therapies, e.g., ASOs to decrease TDP-43 aggregation. Newly emerging biomarkers from EVs containing mitochondrial cargo are becoming useful for guiding mitochondrial-targeted therapies, e.g., PB/TURSO or SS-31 [237]. From clinical trial outcomes, these biomarkers provide an opportunity for clinicians to tailor approaches to therapies to a degree that was impossible before, thus optimizing the therapeutic benefit while minimizing toxicity and side effects for patients [229].

#### 5.1.3. Experimental and Off-Label Drug Applications: Metformin, Tamoxifen, and Ceftriaxone

Numerous repositioned drugs and experimental uses on an off-label basis targeting metabolic stress, protein misfolding, and excitotoxic damage are now being explored in fast progressors for ALS. Metformin as an activator of AMPK has shown metabolic stress reduction as well as autophagy induction and mitigation of certain mitochondrial function [238,239]. It triggers energy-sensing pathways that promote autophagic clearance of protein aggregates and damaged mitochondria, especially pertinent in C9orf72- and TARDBP-related ALS. Analogously, the estrogen receptor modulator, tamoxifen, has been demonstrated to have anti-apoptotic properties in preclinical ALS models through inhibition of caspase-mediated cell death [240].

Ceftriaxone was developed as an antibiotic, but it has garnered attention for its ability to upregulate GLT-1 expression, which is the primary glutamate transporter in the central nervous system. The upregulation of GLT-1 results in increased uptake and clearance of glutamate from the synaptic terminal, helping to diminish synaptic hyperexcitability and excitotoxicity whenever used in conjunction with riluzole or NMDA receptor antagonists [241].

FDA-approved and off-label drug treatments provide a base framework for ALS therapy, but faster progressors must have multimodal and individualized treatment approaches to address their accelerated disease course. With a combination of glutamate inhibitors, antioxidants, mitochondrial-targeting drugs, and RNA-targeted therapies, their new regimen aims to provide neuroprotection in its entirety. Biomarker-guided neuroprotection, including tracking NfL, pTDP-43 diagnoses, and mitochondrial markers, will refine this process as they are quickly or dynamically changed. These analyses should, in theory, launch the understanding of individualized treatment and open a window for precision treatment. Future trials of riluzole, PB/TURSO, edaravone, and future off-label therapies/agents should, in theory, present a viable opportunity clinically to slow motor neuron degeneration and potentially provide, at least, the potential for survival from six months to two years beyond clinical expectations [242].

### 5.2. Gene Therapy and RNA-Based Approaches: Precision Treatments for Targeted ALS Subtypes

Gene therapies and RNA-based options have revolutionized therapeutic options for ALS that target pathogenic gene alterations/variants, toxic RNA foci, and protein aggregation that cause motor neuron degeneration and drive treatment [243]. For the fast progressors, who have a rapid trajectory and need aggressive treatment methods, these approaches should be able to cease progression and slow down the accrual of toxic proteins and correct the gene alteration/mutation. New methods for gene therapies and RNA-based options are being developed at a rapid pace and include ASOs, CRISPR-based genome editing, small-interfering RNAs (siRNAs), and AAV9 delivery. All methods now have progress on specific genes and unique disease-modifying treatment options. Clinical trials of these therapies have already shown success in halting pathology associated with neurodegeneration and stopping functional decline. Provided multimodal combination treatment methods and delivery systems, using biomarker-guided progressions of treatment should bring conceptions about ALS treatment—that are exponentially relevant and make physiological and logical sense—together in the future and result in extending survival for ALS patients, particularly fast progressors [244]. The table below (Table 1) aims to provide an overview of key pathways driving disease progression, the genetic variants involved, and the therapeutic strategies currently under development or clinical investigation.

#### 5.2.1. Tofersen (SOD1-ALS Antisense Therapy): A Groundbreaking Milestone in RNA-Based ALS Treatment

Tofersen is the first ASO developed by Biogen and represents the first potentially targeted RNA therapy to show clinical advancement in advanced stages of clinical trial testing in ALS. Tofersen binds and degrades SOD1 mRNA via prerequisite activation of RNase H, thereby preventing the ability to produce misfolded, neurotoxic SOD1 protein. Mutant SOD1 proteins have been implicated as one of several drivers of familial ALS because they aggregate in toxic ways, depleting energy from mitochondrial function, causing oxidative stress, and inducing neuroinflammation that eventually results in death of motor neurons and accelerated progression of the disease [146].

Tofersen was shown to significantly reduce levels of SOD1 protein as well as levels of NfL concentration, which is known to be a biomarker of axonal degeneration and progression of the disease [263]. In the Phase 3 VALOR trial, early treatment with tofersen demonstrated association with a reduced rate in ALSFRS-R decline, extended survival, and preserved respiratory function. Notably, participants who benefitted most from early treatment of tofersen had a moderate to fast progression of ALS, as subsequent studies are looking at optimal dosing and combinations with other treatment regimens to include alternative products that also work with rapid progressors of the disease [264].

Despite tofersen’s promising clinical outcomes thus far, significantly more could still be done to advance its therapeutic outcome; efforts are in place to develop next-generation ASOs that embody significant chemical modifications to improve stability and CNS penetration, thus adding some target specificity. These modifications include the addition of phosphorothioate backbones, 2′-O-methyl sugar units, and locked nucleic acid (LNA) modifications that may have the ability to increase duration of therapeutic effects, reduce off-target engagements, and reduce immune responses. Combinations of ASOs with mitochondrial protectors such as PB/TURSO or SS-31 might provide a way to simultaneously resolve upstream SOD1 toxicity with a downstream target impacting mitochondrial quality utilizing multiple drugs [265].

#### 5.2.2. C9orf72-Targeted Therapies: ASOs and CRISPR-Based Approaches to Eliminate Toxic RNA and DPRs

The most common genetic cause of ALS is C9orf72 expansions, which are pathological G4C2 hexanucleotide repeats. C9orf72’s direct consequence is toxic RNA foci and the production of dipeptide repeat proteins (DPRs) due to missplicing the gene [266]. The misspliced toxic products have many downstream effects, which alter RNA metabolism, affecting translation and the nuclear capacity to import/export molecules and causing destruction of mitochondrial homeostasis which likely contributes to the fast-progressing form of ALS presented in C9orf72 patients. The future of therapies targeting C9orf72 will aim to target the toxic product either at the source by using ASOs or correcting it with CRISPR-based genome editing methods to treat it in a more permanent manner [267].

ASOs for C9orf72 target the transcript directly by binding to the expanded G4C2 repeat RNA and inducing degradation (RNA “knockdown”)—blocking RNA foci formation and completely removing DPR formation. In preclinical studies of ALS models, C9orf72 ASOs have reduced RNA foci and DPR aggregate formation and improved motor deficits in treated animals—along with strong improvement in mitochondrial function. To date, in clinical trials, early results in ASO-treated patients have been promising, including reduction in relative speed of functional decline and decreased and improved biomarkers of cellular health [268].

The CRISPR-Cas9 gene-editing technology presents a potentially permanent solution to the toxic RNA problem by excising the expanded G4C2 repeats in the genome, completely removing toxic RNA and returning gene expression to normal. In preclinical studies CRISPR-mediated excision improved motor neuron survival, interacted positively with mitochondrial dynamics, and ameliorated oxidative stress caused by toxic RNA. Once therapeutic development is underway, delivery of the CRISPR components will use AAV9 and lipid particles with optimization for CNS targeting in combination to minimize off-target effects [269]. Newly emerging multigene therapy strategies, in which AAV vectors codeliver both CRISPR-Cas9 components and neuroprotective factors (for example, VEGF or BDNF), are being developed to provide the combined effects of genetic correction and protection of motor neurons from loss or dysfunction [270].

Moreover, it is anticipated that C9orf72 ASOs or CRISPR therapies combined with autophagy inducers (e.g., rapamycin or trehalose) will further enhance cellular clearance mechanisms and reduce toxic protein accumulation. The recent developments in C9orf72 ASOs and CRISPR technologies, together with treatment strategies that address the consequences of C9orf72-driven ALS, represent a comprehensive approach to treating both the cause and consequence of C9orf72-driven ALS [165].

#### 5.2.3. FUS-ALS Gene Therapy: AAV9-Based Trials for Rapidly Progressing ALS

FUS variants, particularly the P525L and R521C variant variants, are related to early-onset ALS with accelerated motor decline and poor prognoses [171]. The major pathological feature of FUS-ALS is considered to be the pathological mislocalization of the FUS protein from the nucleus to the cytoplasm as cytoplasmic aggregates, which block normal RNA metabolism and alter the dynamics of the stress granule response and mitochondrial dysfunction. Due to the aggressive nature of FUS-related ALS, gene therapies geared to both decrease the expression of the mutation and restore FUS nuclear localization are critical especially for fast progressors [271].

Given that AAV9-based implementations of gene therapy represent a reasonable platform to introduce either gene-silencing tools or genome-editing components in motor neurons for the treatment of ALS, AAV9 could be useful to deliver gene-silencing or other gene therapies through the blood–brain barrier (BBB) where they can efficiently use CNS target tissue [272]. Target discoveries currently being investigated as gene therapy approaches are:AAV9 delivery of ASOs or siRNAs to silence mutant FUS and reduce cytoplasmic aggregate burden [273].AAV9 delivery of CRISPR-Cas9 for permanent correction of FUS variants, especially gain-of-function variants [274].AAV9 delivery of neuroprotective factors (e.g., VEGF and GDNF) to protect motor neurons from cell death while tackling FUS toxicity [275].

In preclinical models of FUS-ALS, AAV9 delivery of ASOs and CRISPR-Cas9 therapies lower levels of FUS aggregation, restore mitochondrial function, and improve motor performance. Ongoing clinical trials are seeking the best doses, best immune evasion strategies, and best delivery platforms to provide maximal efficacy and long-term safety. Advanced capsid engineering of AAV9 is also occurring to reduce immune response and allow repeat dosing—a major limitation in AAV-based therapies [181].

#### 5.2.4. Innovations in Delivery Systems: AAV Engineering, Lipid Nanoparticles, and Immune Evasion

In order for gene therapies to maximize therapeutic outcomes, efficient and targeted delivery to motor neurons is key. Relative successes in developing engineered AAV capsids and non-viral delivery systems that resolve problems associated with the BBB, ameliorate off-target effects, and evade an immune response are making headway toward resolving specific outcomes among motor neurons and avoiding immune rejection [276].

Engineered AAV capsids, such as AAV-PHP.eB and its other next-generation variants, have favorable tropism for motor neurons and improved CNS delivery, while also being engineered to evade neutralizing antibodies to allow for repeated doses without inducing an immune response. Non-viral delivery systems, such as lipid nanoparticles, are poised to solve the problems that AAVs have with immune responses and affinity for target neurons and to remedy immunogenicity with delivery. Self-amplifying RNA systems to improve gene expression efficiency and delivery systems employed as adjuvants are also being examined to lessen the number of doses needed to achieve therapeutic efficacies for RNA-based therapeutics like ASOs or siRNA [277].

#### 5.2.5. Biomarker-Guided Optimization: Personalizing Gene Therapy for Fast Progressors

Biomarkers will be required to make RNA-based and gene-editing therapies viable options, including the ability to optimize treatment-related regimens and timely monitor treatment response. Various biomarkers (NfL) now exist for CSF and plasma to measure axonal degeneration and treatment-related response. Reduced NfL levels are consistent with treatment response to declining axonal depression as seen with intrathecal tofersen. Further, pTDP-43 and stress-granule-associated proteins can be used to monitor related therapeutic success for FUS- and TARDBP-targeted therapies, while mitochondrial biomarkers/features (differences in ROS levels or mitochondrial membrane potential) can be applied to monitor combination therapies using mitochondrial protectors with RNA-based therapies, etc. [278].

### 5.3. Stem Cell and Regenerative Therapies: Restoring and Protecting Motor Neurons in Fast Progressors

Stem cell and regenerative therapies are also a revolutionary means of treating fast ALS progressors by addressing irreversible motor neuron loss and creating an environment that encourages neural regeneration and survival. Instead of just delaying disease progression, regenerative therapies work to protect motor neurons, repair axons, and regenerate synapses [279]. Two of the most effective regenerative therapies—NurOwn (mesenchymal stem cell therapy) and neural stem cell (NSC) transplantation—have shown promise in delivering neurotrophic support, modulating neuroinflammation, and restoring functional motor networks [280]. Advances in genetic engineering and exosome therapy, along with the development of biomaterial scaffolds, will help to improve the durability and effectiveness of regenerative therapies and set the stage for personalized regenerative medicine in ALS [280].

#### 5.3.1. Mechanisms of Action: Beyond Neurotrophic Support

Stem cell-based therapies provide neuroprotection via a number of different mechanisms, as well as secretion of neurotrophic factors, that target numerous pathways involved in motor neuron degeneration. NurOwn, an autologous MSC therapy, increases motor neuron survival through the secretion of a mixture of growth factors, such as brain-derived neurotrophic factor (BDNF), glial-cell-line-derived neurotrophic factor (GDNF), insulin like growth factor 1 (IGF-1), and hepatocyte growth factor (HGF) [281]. These neurotrophic growth factors not only support motor neuron protection and growth but also increase axonal regeneration, preserve synaptic connections, and facilitate neurofilament stability [282].

In addition to neurotrophic support from MSC or NSC therapy, these cells also offer mitochondrial protection by restoring mitochondrial membrane potential and stimulating ATP production. Growth factors like IGF-1 and VEGF induce mitochondrial biogenesis, inhibiting ROS production and mitochondrial swelling, which are important to prevent energy deficits associated with rapidly progressing ALS. In addition to initiating mitochondrial biogenesis, neurotrophic factors affect axonal transport by stabilizing microtubule networks and promoting efficient transport of mitochondria and synaptic vesicles, which will deter or at least delay functional deficits of neurons and provoke axonal degeneration. Both BDNF and IGF-1 also enhance synaptic remodeling and synaptic plasticity, stimulating synaptogenesis to create new functional connections between the neurons in the synapse [283].

#### 5.3.2. NurOwn (Mesenchymal Stem Cells—MSCs): Secreting Neurotrophic and Anti-Inflammatory Factors

In terms of neurotrophic and anti-inflammatory factors, NurOwn, developed by BrainStorm Cell Therapeutics, is one of the most advanced MSC-based therapies in the ALS clinical trial pipeline. MSCs are isolated from the patient’s bone marrow, expanded, and subsequently induced to secrete high levels of neurotrophic factors before being reintroduced to the patient via intrathecal injection directly into the cerebrospinal fluid. In addition to neurotrophic factors, NurOwn-modified MSCs exhibit powerful anti-inflammatory effects by secreting anti-inflammatory cytokines (e.g., IL-10) and inhibiting release of pro-inflammatory mediators (e.g., IL-1β and TNF-α). This modification of the inflammatory microenvironment significantly reduces the neurotoxic effects attributed to chronic microglial activation associated with rapidly progressing ALS [284].

Clinical trial data of NurOwn show fairly heterogeneous but ultimately hopeful findings. In responders, significant improvements in muscle strength and respiratory function and a decrease in NfL levels, which is consistent with reduced axonal damage and motor decline, were observed. However, it is apparent that different outcomes will be observed among responders and/or better defined groups of ALS cases, highlighting the need for biomarker-guided patient stratification by specific subgroups (i.e., milder disease, early stage disease) estimated to benefit the most from NurOwn therapy [285]. Future studies will evaluate combinations of NurOwn with adjunctive therapies, such as mitochondrial stabilizer (e.g., SS-31) and RNA-directed therapies, to optimize performance in faster progressors [286].

#### 5.3.3. Neural Stem Cell Therapy: Replacing Lost Motor Neurons and Rebuilding Neural Circuits

NSC therapy offers the potential to reduce lost motor neurons and reestablish motor circuits via the transplantation of progenitor cells that can differentiate into a true neuron. NSCs are different than MSCs which provide neurotrophic support primarily but do not typically differentiate into functioning motor neurons with the capacity to connect existing networks. NSCs also provide secretion of neurotrophic factor, promote synaptic growth, and facilitate axonal regeneration, giving neuroprotection and structural repair [287].

NSCs, including spinal cord-derived NPCs, have been investigated in clinical trials and have demonstrated effectiveness and long-term outcomes suggestive of cellular survival, axonal extension, and improved motor function; and several studies have provided evidence that NSC transplants into ALS patients’ spinal cords were safe and that the decline in clinical function of some study participants was less than the decline in previous stages before the transplant [288]. Yet, we still face hurdles with respect to sustainability of the graft, integration functionality, and achieving immune tolerance. To alleviate these challenges, engineers are working towards creating NSCs with a genetic component (eNSCs) to produce anti-apoptotic proteins, some hopeful examples including Bcl-2 proteins, and neurotrophic factors such as VEGF or GDNF in order to optimize both sustainability and efficacy when performing transplants [289].

Researchers are exploring gene-modified NSCs for delivery of CRISPR-Cas9 or ASO-based gene editing tools to motor neurons in situ to correct genetic variants (i.e., SOD1, C9orf72, or TARDBP). Gene-edited NSCs are the best chance of integrating neuroregeneration with sustainable genetic correction, therefore making them particularly compelling candidate cells for currently identified genetic variant carriers [290].

#### 5.3.4. Exosome-Based Stem Cell Therapies: A Cell-Free Alternative

Exosomes derived from MSCs or NSCs are starting to be considered as a natural cell-free counterpart to stem cell therapies and could provide an exciting therapeutic option for patients. These nanoscale EVs offer a potential cargo of neurotrophic factors, anti-inflammatory cytokines, miRNAs, and mitochondrial-regulatory proteins acting to provide neuroprotection, synaptic repair, and anti-inflammatory signaling [291]. Exosomes can avoid many of the barriers present in standard stem cell transplants, including immune rejection and limiting cell survival, and can offer an easily accessible method of delivering therapeutic agents that failed by other means, preferably by intravenous or intrathecal injections [292].

Preclinical studies with MSC exosomes showed that treatment reduced neuroinflammation, provided neuroprotection against mitochondrial dysfunction, as well as supported neuronal survival in ALS models (i.e., mice). There is also combined treatment with similar mitochondrial protection agents (i.e., SS31) and nucleic-acid-based therapies being developed to improve their neuroprotective potential [293].

#### 5.3.5. Biomaterials and Scaffolds to Improve Stem Cell Integration

In order to replace damaged motor circuits one must first create an optimal environment to increase cell survival and functional differentiation. Biomaterials such as scaffolding, hydrogels, and ECM-based structures are also being developed to facilitate cell attachment, protect cells against apoptosis, and promote synaptic integration. Biomaterials as scaffolds can provide structural support, by mimicking the native ECM and delivering neurotrophic factors to support axonal growth pathways and synaptic repair. The authors used hydrogels containing MSCs and demonstrated improvements in cell attachment and retention, as well as functional efficacy in preclinical ALS models. The utilization of biomaterials along with stem cells may help maximize the long-term effects and durability of induced regenerative therapies [294].

#### 5.3.6. Biomarker-Guided Monitoring of Stem Cell Therapies

The use of advanced biomarkers will be essential to augment the operational aspects of stem cell therapies, by providing real-time, continuous monitoring of both treatment responses and cell survival and integration into functional motor circuits. Amongst the more commonly used biomarkers of axonal degeneration, the one that is most often used to measure the therapeutic advantages of regenerative therapies is NfL [295]. EVs from transplanted islets themselves could also be an alternative to therapeutic effects in vivo and/or grafting. Specific imaging modalities may be able to non-invasively image stem cell migration (i.e., bioluminescent) or integration (i.e., MRI-based tracking). Modified cytokine profiles via circulating serum cytokines, for example, decreased IL-1β and increased levels of IL-10, may further indicate potentially orchestrating an, ironically, anti-inflammatory state in the nervous system and correlate to target functions of physiological significance or resolution [296].

Stem cells and regenerative therapies are fundamentally multifactorial strategies to achieve neuroprotective and neural regenerative treatment of rapidly progressive ALS. The relative speed with which NurOwn (neurotrophic and anti-inflammatory) acts excludes NSCs from the capacity of organ replacement but provides the most optimism in in terms of trying to address the CPG and motor neuron replacement. New emerging technologies and innovation (e.g., exosome-based therapies, gene-edited stem cells, and biomaterials or scaffolds) are entirely new technologies for enhancing stem cells and their integration and ultimately enhancing their efficacy potential for neuronal delivery. Support strategies such as mitochondrial stabilizers, autophagy inducers, RNA-targeted therapies, etc. may also be developed as enhancements alongside regenerative therapies. Presumably, with a more complete strategy, we may still slow the disease progression and offer some real advantage for these correspondingly rapid progression ALS patients [297].

### 5.4. Neuroprotective and Anti-Inflammatory Therapies: Targeting Cellular Stress and Chronic Inflammation in Fast ALS Progressors

Chronic neuroinflammation and dysfunctional cellular stress responses are core features of fast ALS progressors, resulting in rapid motoneuron loss. Dysregulated microglia, astrocytes, and peripheral immune cells are activated and release pro-inflammatory cytokines such as IL-6, TNF-α, IL-1β, and CXCL10, leading to oxidative stress, mitochondrial dysfunction, and synaptogenic loss [298]. Concurrently, impaired proteostasis, including dysfunction in autophagocytosis, leads to aggregation and accumulation of misfolded proteins such as TDP-43 aggregates, SOD1 aggregates, and dipeptide repeat proteins that further exacerbate motoneuron degeneration. The emerging neuroprotective therapies outlined in this section seek to modify inflammation, enhance autophagy, and restore proteostasis with one potential therapy being a multifaceted control of the progression of ALS [299]. This section aims to focus on tocilizumab and other cytokine inhibitors, mTOR modulators, heat shock protein inducers, and new modalities of therapy targeting the unfolded protein response (UPR) that target the aggressive nature of ALS disease in fast progressor patients.

#### 5.4.1. IL-6 and TNF-α Blockers: Suppressing Cytokine Cascades and Protecting Motor Neurons

Pro-inflammatory cytokines (IL-6, TNF-α, and IL-1β) are significant mediators of neuroinflammation, accelerating mitochondrial dysfunction, ROS production, and subsequent apoptosis of motoneurons. Elevated CSF and plasma cytokine levels have been associated with faster ALS progression, greater oxidative damage, and worse clinical outcomes due to already existing inflammation. Fast ALS progressors are more susceptible to damage driven by pro-inflammatory cytokines, due to their inherently higher inflammatory state in the brain tissue [300,301].

Tocilizumab is an IL-6 receptor blocker that has demonstrated significant effectiveness in reducing IL-6-mediated activation of microglia and protecting motoneurons from cell death due to inflammatory damage. By blocking IL-6 signaling, tocilizumab blocks the downstream release of ROS and pro-apoptotic signals while maintaining motoneuron viability. Early investigations have shown that some patients with ALS have demonstrated stabilization of muscle strength and a reduction in biomarkers of neurodegeneration while on treatment. New research is investigating whether it can be used in combination with mitochondrial-targeting compounds (for example, PB/TURSO or SS-31) for ALS specifically to mitigate inflammatory and metabolic dysfunction [302].

XPro1595 is a selective TNF-α signaling antagonist that blocks the soluble neurotoxic TNF-α and spares transmembrane TNF-α, since transmembrane TNF-α is required for immune homeostasis. In preclinical studies using several ALS mouse models, XPro1595 treatment produced significant inhibitions on microglial activation, preservation of synaptic integrity, and overall improved speed and endurance of the forelimb. In summary, XPro1595 abolishes TNF-α signaling neurotoxicity, abolishes excessively reactive neuroinflammation, and suppresses the resultant ROS neurotoxic effects, leading to motor neuron death. Future plans are to study XPro1595 in combination with RNA-targeting therapies (for example, tofersen (SOD1 ASOs) or ASOs targeting C9orf72) to achieve a combined effect of inhibiting neuroinflammation in addition to inhibiting protein-folding and proteotoxic stress associated with ALS [303].

There are other inflammatory targets, such as IL-1β, IL-18, and CXCL10 (which also mediates inflammation), which have all been examined as to their potential contribution to potentiating an inflammatory response. Once again, the NLRP3 inflammasome is activated to facilitate proteolytic processing to obtain pro-IL-1β; this gating event activates some mitochondrial insult and inhibits the disposal of damaged proteins. An NLRP3 inhibitor, such as MCC950, that is known to block IL-1β induction, could be purposed to unravel the effects of mitochondrial permeability transition and both major deficits on motor function in SOD1-ALS models. CXCL10 mediates microglial recruitment, and once CXCR3-mediated microglial recruitment is inhibited, there was no proximal inflammation from the ALS lesion [304].

#### 5.4.2. Rapamycin and mTOR Modulators: Enhancing Autophagy and Mitigating Proteotoxic Stress

Degeneration of the neuromuscular system and neuroinflammation are key features of ALS, in particular the faster progressing forms of ALS. For example, these issues are noteworthy during peripheral nerve involvement, since toxic protein inclusions occur. Examples of toxic protein inclusions associated with proteotoxic stress are TDP-43, SOD1 aggregates, and dipeptide repeat proteins (DPRs). Dysregulated autophagy and neuroinflammation are some of the key core causes of motor neuron degeneration in ALS. In the event of proteotoxicity, the autophagic activity of the cell, the body’s natural degradation system utilized to clear damaged organelles or protein aggregates, is either incapable or limited in clearing the misfolded proteins. Notably, this defective autophagic flux in the accumulation and toxicity of substances can either induce, or be a potential cause of, pathogenic motor neuron degeneration in ALS. Long-term defective autophagic flux in motor neurons can cease to clear toxic, permanent inclusions (above), leading to substantial deposition of protein aggregates and loss of cellular homeostasis [73].

Rapamycin, an mTORC1 inhibitor, works primarily to promote autophagy by inhibiting the mechanistic target of rapamycin (mTOR), which acts as a central controller of cellular growth and thereby protein synthesis. Increasing autophagic activity via rapamycin improves the clearance of misfolded aggregates, overall decreasing proteotoxic stress and restoring cellular homeostasis. In preclinical models, rapamycin has demonstrated the ability to decrease the accumulation of both TDP-43 and SOD1 aggregates, improve mitochondrial function, and decrease motor dysfunction. In particular, in C9orf72 models of ALS, rapamycin was able to better clear consequences of DPRs, allowing their degradation. Decreased accumulation of DPRs prevented transport across the nuclear envelope, while also impacting mitochondrial function [305].

However, long-term mTORC1 inhibition via rapamycin may become an issue suppressing protein synthesis and maintenance of the cytoskeleton. To counteract unwanted effects of chronic inhibition of mTORC1, on mTORC2 that regulates the cytoskeleton and, ultimately, neuron survival, dual mode mTOR modulators have been produced [305]. Additionally, there are several small molecules that have been generated which target the transcription factor EB (TFEB), a master regulator of lysosomal biogenesis. These new compounds should facilitate autophagic flux and lysosomal capacity for degradation of substrates. It is possible that a combination of rapamycin with TFEB activators, chemicals that induce heat shock proteins like arimoclomol, may induce additional synergistic effects in promoting clearance of toxic aggregates [306].

#### 5.4.3. Arimoclomol: Heat Shock Protein Modulation to Stabilize Misfolded Proteins

Arimoclomol may be a pharmacological inducer of heat shock proteins (HSPs). It enables cellular competence and quality, as well as folding, mitigating, and preventing aggregation of proteins. In compromised ALS cells, HSPs are depleted, resulting in misfolded or aggregated toxic proteins, such as TDP-43 and SOD1. Arimoclomol selectively upregulates expression of HSP70, HSP90, and other stress-induced chaperones which allows the cell to refold or degrade misfolded proteins. The positive effects of arimoclomol can be posited as a relational change in the cellular dynamic influencing proteostasis as well as the state of the cell. From preclinical evidence, it appears that Namenda reduces MTB aggregation, clears mitochondrial decay, and prolongs motor function in animal models of ALS [307].

In human clinical trial research, arimoclomol is shown to improve functional outcomes and is well-tolerated in patients with SOD1 variants [308]. It is therefore plausible to consider use of arimoclomol along with any other treatments. For example, if the coadministration of arimoclomol and rapamycin allows for protein clearance to occur through both autophagic degradation and chaperone-assisted refolding. Arimoclomol may also be considered along with potential mitochondria-targeted neuroprotection outcomes (e.g., edaravone or SS-31) such that any mitochondria proteins found in the studies may be stabilized and the chance of protection from oxidative stress augmented [309].

#### 5.4.4. Targeting ER Stress and the Unfolded Protein Response

ER stress and the UPR are likely the largest contributory factors to the motor neuron degeneration seen in ALS. After the overload of misfolded proteins that is beyond the folding amount of the ER, the ER stress response results in the activation of the PERK pathway that results in overexcitation, inactivation of translation, and activation of apoptosis. Inhibition of UPR pathways provides another layer of therapeutic intervention by reducing ER stress and enhancing the degradation of misfolded proteins [310].

PERK inhibitors (e.g., ISRIB) that inhibit overstimulated PERK activity will restore protein-translational capacity and therefore preserve synaptic maintenance and motor neuron function, while IRE1α RNase inhibitors reduce XBP1 splicing and decrease pro-apoptotic activity, and ATF6 activators increase expression of ER chaperones, allowing protein refolding and reducing aggregate accumulation. Combination therapies that involve UPR inhibitors and autophagy inducers may have synergistic effects in the reduction of both ER-stress-induced apoptosis and proteotoxic burden [311]. The figure below (Figure 3) intends to illustrate how the dysregulated PERK pathway contributes to motor neuron degeneration through disruptions in these essential cellular processes.

#### 5.4.5. Biomarker-Guided Optimization of Anti-Inflammatory and Neuroprotective Therapies

Future biomarker development is needed for effective treatment plans and dynamic follow-up of treatment response. For example, the last set of cytokine value profiling (i.e., IL-6, TNF-α, IL-1β) in CSF and plasma can be used to determine cytokine inhibitors (e.g., tocilizumab, XPro1595) based on the inflammatory state of the patient. Mitochondrial biomarkers (including ROS levels, ATP production, and mitochondrial membrane potential) will also be critical to optimize combination regimens that involve anti-inflammatory agents, along with the mitochondrial protectors, discussed previously. Reductions in pTDP-43 aggregates in CSF or EVs will act as surrogate markers for better protein clearance after treatments involving both autophagy enhancement and induction of heat shock proteins [312].

Neuroprotective and anti-inflammatory drugs will be important to slow the rapid progression of ALS in fast progressors due to targeting of interconnected mechanisms of inflammation, proteotoxicity, and cell stress. IL-6 and TNF-α blockers will reduce cascading neuroinflammatory pathways. Therapeutics, such as rapamycin and dual mTOR modulators will restore autophagic homeostasis to clear toxic aggregates. Arimoclomol will facilitate cleavage-mediated stability through heat shock protein induction and new UPR inhibitors add an innovative mechanism in mitigating ER stress. In multimodal therapies, these strategies will provide tremendous neuroprotection, improve survival, and improve quality of life for patients. Future personalized interventions and biomarker-guided methodologies for fast progressors are really the future of determining optimal therapeutic applications for their life-limiting disease progression at this time [313].

### 5.5. Metabolic and Nutritional Strategies: Enhancing Energy Balance and Mitochondrial Function in Fast ALS Progressors

Metabolic dysfunction and energy deficits are key aspects of ALS, particularly in fast- progressing ALS patients who are thought to develop hypermetabolism, rapid weight loss, and impaired mitochondrial function. These metabolic dysfunctions, known as hypermetabolism, accelerate motor neuron death by increasing oxidative stress, depleting ATP levels, and causing a systemic catabolic state [314]. A variety of metabolic and nutritional strategies that include calorically rich diets, ketogenic diets, and mitochondrial-targeting drugs such as AMX0035 (Relyvrio) are potential therapies aimed at restoring energy homoeostasis, improving mitochondrial function, and slowing ALS progression. The growing understanding of novel and emerging ways to measure, understand, and modulate metabolism such as the gut–brain axis, harnessing brown adipose tissue, and effectively using multiomics approaches can lead to ways to personalize interventions for individuals with different metabolic presentations of ALS [315].

#### 5.5.1. High-Calorie, High-Fat Diet: Countering Hypermetabolism and Extending Survival

Increased energy demands seen in fast-progressing ALS, often termed hypermetabolism, can lead to 30–50% more body energy expenditure than healthy controls due to having increased metabolic demands and a chronic inflammatory state. This leads to an imbalance of energy levels, depletion of skeletal muscle and fat mass, and a negative energy status for ALS patients. Fast-progressing ALS studies have demonstrated that all of these metabolic changes are linked to faster disease progression and shorter survival times. High-calorie, high-fat diets are designed to address these issues and to correct the caloric deficit by providing an abundant source of calories, mostly in the form of lipids and utilizing lipids can be up to 2–3× more effective for providing energy than carbohydrates [316].

Interestingly, clinical studies have shown that ALS patients who can maintain or increase their body weight even by using high-calorie diets had improved survival rates. The most timely illustrative study showed that fast-progressing ALS patients who ate high-calorie diets with medium-chain triglycerides (MCTs) or omega-3 fatty acids had slower functional decline, longer survival, and improved mitochondrial function. Lipid metabolism generates higher amounts of ATP than carbohydrates per gram of substrate, which reduces metabolic stress and deemphasizes glycolytic pathways that are often dysfunctional in ALS [317].

Omega-3 fatty acids (i.e., EPA and DHA) additionally provide neuroprotective aspects of reducing neuroinflammation, maintaining synaptic integrity, and stabilizing neuronal membranes. In addition, the anti-inflammatory properties of omega-3 fatty acids will modulate the activation of microglial cells and the release of cytokines, which are drivers of motor neuron degeneration. Emerging combination strategies are being evaluated that pair high-caloric diets with mitochondrial-targeting therapies (i.e., AMX0035 (Relyvrio) or SS-31) to address energy deficits due to the inefficiencies of glycolytic pathways and mitochondrial dysfunction [318].

Multinutrient formulations containing MCT oil, antioxidants, and protein supplements are being developed to improve muscle mass and fat utilization and reduce oxidative stress. Furthermore, individual adjustments made using resting energy expenditure (REE) measurements can ensure proper caloric intake individualized based on assessment [319].

#### 5.5.2. The Ketogenic Diet and Exogenous Ketone Supplements: Neuroprotection Through Alternative Energy Sources

The ketogenic diet (KD) is a form of dietary therapy that involves high fat, low carbohydrates and moderate dietary protein intake. The purpose of the KD is to induce ketosis in which the body produces ketone bodies (e.g., β-hydroxybutyrate (BHB) and acetoacetate) as an alternative energy source to glucose. This is advantageous in ALS, since neurons preferentially metabolize ketones, and ketones bypass glucose-dependent metabolic pathways, are more efficient in generating ATP, and produce less ROS than glucose [320]. Preclinical models of ALS have demonstrated that the ketogenic diet reduces motor neuron loss, improves mitochondrial health and function, and reduces markers of neuroinflammation. Ketones improve mitochondrial membrane potential and inhibit apoptosis related to mitochondrial outer membrane permeabilization by reducing oxidative stress and preventing the mitochondrial permeability transition pore (mPTP) from opening. Furthermore, BHB has been evidenced to decrease NLRP3 inflammasome signaling which is a major mediator of chronic neuroinflammation in ALS patients [321].

Even with a positive therapeutic profile of the high-fat ketogenic diet in ALS patients, it is still limited in clinical practice due to compliance and GI-adverse effects of dietary change. Considering these limitations, exogenous ketones such as ketone esters and ketone salts are of interest as potential alternative strategies for clinical ketosis. Exogenous ketones can provide therapeutic adaptations while minimizing total dietary changes. Exogenous ketones allow for fast and sustained ketosis easily and with less dietary control than dietary ketosis. Exogenous ketones are also gaining interest with the application of autophagy inducers (e.g., rapamycin or trehalose) to facilitate protein clearance and provide rapid or alternative energy sources in rapid progressors with ALS [322].

#### 5.5.3. The Gut–Brain Axis: A Novel Metabolic Target in ALS Therapy

Emerging evidence highlights the gut–brain axis as an important part of ALS disease progression and connects gut dysbiosis (i.e., altered gut microbes) to systemic inflammation, disrupted energy metabolism, and neurodegeneration. For instance, patients with ALS (including fast progressors) exhibit changes in the compositional diversity of gut microbiota and elevated gut permeability (i.e., “leaky gut syndrome”), resulting in the leakage of pro-inflammatory cytokines and metabolites into systemic circulation [323].

Probiotics to restore gut homoeostasis, resulting in enhanced production of short-chain fatty acids (SCFAs). Particularly, butyrate is one of the more attractive treatment modalities. Butyrate in particular supports mitochondrial respiration, augments ATP production, and upregulates antioxidant defenses via inhibition of histone deacetylases (HDACs). Probiotics that generate SCFAs, combined with a high-calorie/ketogenic diet, may have dual benefit of reducing systemic inflammation while also improving mitochondrial function [324].

Moreover, it is also worth noting that microbial modulation of the kynurenine pathway is a regulator of NAD+ biosynthesis; NAD+ is a key cofactor of mitochondrial function and sirtuin activation. Personalized gut-based interventions based on microbiome sequencing and metabolic profiling are likely to maximize probiotic and prebiotic supplementation based on individual gut profiles [325].

#### 5.5.4. Brown Adipose Tissue (BAT) Activation: A Novel Energy-Regulating Mechanism

Brown adipose tissue (BAT) is a tissue essential for non-shivering thermogenesis and fat and energy metabolism and uses stored fat, converting it into heat. Due to the anticipated role of BAT activation as a therapeutic avenue to offset hypermetabolism (which is commonly seen in ALS) and energy deficiencies, incorporating BAT activation as a metabolic therapy seems plausible. BAT may be activated through cold exposure (thermogenic activation), exercise (which activates BAT), and through beta3-adrenergic agonists such as mirabegron. BAT activation may lead to increased fat oxidation, mitochondrial biogenesis, and systemic availability of energy reserves and has the potential to change energy flux in muscular tissues. BAT also secretes batokines like FGF21, with nutritional responsiveness (improved muscle metabolic function) and insulin sensitivity. The therapeutic benefits of BAT activation may compliment the use of fatty and high-caloric diets or decoupling with light or ketogenic dietary approaches since they support energy use for ATP production to support metabolic health for the brain and muscles in ALS [326].

#### 5.5.5. Biomarker-Guided Personalized Metabolic Interventions

Advanced biomarkers can support individuals’ metabolic therapy, safety, and tolerability for their specific needs.

Monitoring REE and body composition helps recalibrate caloric intake for weight loss and maintaining energy and avoiding muscle loss and negative energy status [327].Plasma ketones evaluate the effectiveness a ketogenic diet or supplemental exogenous ketone in hypermetabolic patients with low energy status and maintenance of energy balance [322].Mitochondrial biomarkers (i.e., ATP production rate or levels of reactive oxygen species, mitochondrial membrane potential) can help assess the impact of the amended mitochondrial targeted therapy [328].Gut microbiome profiling can guide our choice in prebiotic and probiotic supplements after identifying the microbial imbalances contributing to metabolic deficits which may impact ALS and mitochondrial function [329].

Metabolic function and nutrition are pivotal in recognizing the extent of energy deficits and mitochondrial dysfunction underpinning the underlying disease state, particularly with respect to rapidly progressing ALS. High-calorie and ketogenic diets can provide critically needed caloric support, while food stability creates a positive push toward energy balance and the ultimate reduction of oxidative damage. Questions remain regarding the potential of brain–gut axis engagement in metabolic derangement, BAT activation, and individual manipulation of multinutrient formulations. As biomarker-guided and individualized approaches become more commonplace we would expect to see gradual metabolic treatments that extend life and quality of life after BAT activation and planning disability for metabolically impaired clients, particularly in fast progressors [321].

### 5.6. Symptom Management for Fast Progressors: Optimizing Quality of Life and Functional Independence

Symptom management is central to comprehensive ALS care, particularly symptoms and management related to ALS, for fast progressors who experience rapid decreases in respiratory, bulbar, and other functions. While disease-modifying therapies would focus on delaying the changes we see in fast progressors, managing symptoms addresses the complications we face in respect of what matters most for survival and quality of life: respiratory support, nutrition, spasticity and pain, drooling, and augmentative communication technologies. Collectively, through innovations, personalized approaches, and a team-based approach to care, symptom management can reduce complications, increase comfort, and increase survival [236].

#### 5.6.1. Respiratory Support: Expanding Options Beyond Non-Invasive Ventilation

Respiratory failure is the most common cause of death in ALS, particularly fast progressors, who rapidly develop weakness in the diaphragm, experience increased lung intolerance, and develop nocturnal hypoventilation. NIV is the gold standard for managing respiratory insufficiency, as it assists respiratory function with a mask interface and non-invasive approach to support ventilation. Research shows that initiating NIV early—usually when FVC is 50% or less—has been shown to prolong survival by 7 to 12 months, and it can minimize the respiratory symptoms associated with ALS (dyspnea, orthopnea), enabling better sleep quality and functioning. Overall, responses to any ventilation questions can be guided by monitoring—such as polysomnography and overnight oximetry—and make it possible to maximize respiratory support in a personalized way [330].

Various respiratory interventions are in development for implementation to augment, or replace, NIV in situations in which it is insufficient because of bulbar dysfunction or progressive changes with the disease. Diaphragm-pacing systems, which stimulate the phrenic nerve, to maintain persistence in diaphragm contractions, show promise in delaying respiratory decline. Neuromuscular electrical stimulation (NMES) is being studied as a non-invasive way to build strength in respiratory muscles, improve lung function, and prevent atrophy of the diaphragm. High-frequency chest wall oscillation (HFCWO) has also been successfully used to help with airway clearance in patients that have weak cough reflexes, and, while it is possible that it helps prevent respiratory infections via mobilizing secretions, that has not been directly studied in this population. Further, if NMES, HFCWO, and other emerging therapies can be paired with biomarker-guided monitoring of respiratory function, quality of life may be more readily enhanced with individual patient care and adaptive response and eventually prolong survival [331].

#### 5.6.2. Nutritional Support: Gastrostomy Tube Placement and Biomarker-Guided Nutrition

The importance of ensuring adequate nutrition cannot be overstated in ALS, as malnutrition and weight loss indicate rapid progression of the disease and/or reduce survival. In fast progressors, dysphagia (swallowing difficulties) is fairly common and often leads to significant decreases in caloric intake, aspiration risk, and muscle wasting with the associated loss of strength that leads to malnutrition and weight loss. Ideally, placement of a percutaneous endoscopic gastrostomy (PEG) tube or nasogastric (NG) tube will occur when oral intake is no longer sufficient. Importantly, results have shown that PEG tubes could be placed earlier, prior to a patient having an FVC of less than 50%, in order to reduce the risks of aspiration-related complications; improve caloric intake; and improve survival [332].

Nutritional approaches that can be tailored to individual metabolic needs are evolving in the field of ALS care. It is important that we look at metabolic biomarkers, like REE, plasma prealbumin, and C-reactive protein (CRP), to determine caloric intake and recommendations on nutrient makeup. Some enteral tube feeding formulas that are high calorie and high fat with the inclusion of MCT and omega-3 fatty acids potentially have a neuroprotective effect and are a calorically efficient source of calories to counteract hypermetabolism. To that end, we are designing personalized metabolic interventions for patients that include multinutrient formulations with MCTs, omega-3s, and antioxidants to improve energy balance and muscle mass maintenance and minimize oxidation. These individualized approaches are expected to complement mitochondrial-targeting strategies, including ketogenic diets, to deliver a comprehensive metabolic support strategy [333].

#### 5.6.3. Spasticity and Cramps: Combining Pharmacological and Non-Pharmacological Interventions

Spasticity and muscle cramps, due to upper motor neuron hyperexcitability and lower motor neuron ineffectiveness, can cause pain, stiffness, and limitations in mobility and are not uncommon symptoms in fast-progressing patients with ALS. Baclofen is the first-line pharmacological treatment to decrease spasticity by inhibiting spinal reflexes as a GABA-B receptor agonist [334]. Tizanidine is an alpha-2 adrenergic agonist and is more suited to patients with spasticity with pain or sleeping issues. Gabapentin is often used for muscle cramps, and neuropathic pain as well, with gabapentin also lessening unacceptable firing of neurons and causing less muscle spasming [335].

Non-pharmacological approaches are increasingly being recognized as helpful adjuncts or alternatives to drug therapy. Functional electrical stimulation (FES) is a modality that delivers electrical pulses to allow spastic muscles to be relaxed and has been shown to promote muscle extensibility and to improve range of motion and reduce muscle atrophy. Transcranial magnetic stimulation (TMS) is a form of neuromodulation that can selectively modulate cortical motor excitability in order to reduce spasticity. Low-impact physical therapies like hydrotherapy and aquatic exercises can relieve muscle stiffness and improve joint mobility while giving patients a safe non-invasive way to achieve long-term symptom relief. A combined pharmacological and non-pharmacological approach is likely to produce better functional outcomes and improved quality of life [336].

#### 5.6.4. Sialorrhea (Drooling): Advanced Pharmacological and Minimally Invasive Interventions

Sialorrhea, or drooling, is a distressing symptom of ALS driven by loss of oral clearance and impaired swallowing. When neglected, the problem can evolve into choking, aspiration pneumonia, or social isolation. Anticholinergic medications glycopyrrolate and atropine are typically used to decrease salivation by blocking muscarinic receptors, but glycopyrrolate is preferred due to its restricted adverse effects on the central nervous system; to improve salivation, botulinum toxin (Botox) directly injected into the parotid or submandibular glands may provide relief for up to 6 months because it inhibits the release of acetylcholine [337].

Future technologies are altering the treatment of sialorrhea with greater accuracy and efficacy. High-frequency ultrasound-guided botulinum toxin injections can also provide longer-lasting symptom control with greater accuracy. Implantable salivary diversion devices can potentially redirect excessive saliva so as to prevent drooling, while not necessarily reducing total salivation. Portable intraoral suction devices remove saliva in real time, which can continuously be performed in patients with significant or refractory sialorrhea. With specific severity and responses to treatment in mind, the multidisciplinary teams keep track of the treatment responses and tailor their treatment in order to balance efficacy, detection, and side-effects [338].

#### 5.6.5. Assistive Communication: Eye-Tracking Systems and Brain–Computer Interfaces (BCIs)

Many ALS patients will ultimately suffer from significant speech impairment due to bulbar muscle weakness. These conditions impede a patient’s ability to verbally communicate and use speech to accomplish personal, recreational, and professional communicative needs. Eye-tracking systems, such as Tobii Dynavox, have been adopted as a quintessential addition to communicating hands-free, allowing patients to generate speech or text when controlling a device through sequential or coded eye-tracking movements. These systems are highly customizable systems can be linked with home automation, the internet, and environmental control tech, supporting patients in maintaining in-dependence and social engagement [339].

Where patients transition to complete paralysis (locked-in state), BCIs promise to be a truly revolutionary development for assistive communication for these patients. BCIs, based on electroencephalography (EEG), decode brain signals and convert them to digital output, which provides patients an opportunity to communicate or access external devices using only brain activity. And hybrid systems can combine eye tracking with BCIs, providing the most flexibility as disease progresses. Furthermore, BCIs are under development for environmental control; controlling wheelchairs, speech synthesis, and accessing digital devices. As BCI technology progresses, the communication modalities and quality of life for patients with advanced ALS are expected to receive an innovative boost [340].

#### 5.6.6. Emerging Innovations: Personalized and Adaptive Care

As ALS care develops, personalized and adaptive methods of symptom management are also being developed using real-time monitoring, wearable sensors, and artificial-intelligence-powered predictive models to monitor the rate of progression in the disease and adjusting interventions. For example, wearable devices can measure spasticity, as well as cramps and respiratory function, allowing clinicians to make dynamic adjustments to medications and therapy intensity. Also, AI is being used to predict nutritional requirements, that is based on metabolic data, which can help as clinicians develop dietary and PEG tube feeding plans. Emerging developments will allow us to offer more precise, responsive, and personalized health care to optimize outcomes and comfort [12].

Symptom management plays a key role in care for ALS fast progressors that focuses on reducing respiratory distress, nutritional deficits, motor dysfunction, and communication barriers. As above, NIV is generally regarded as the best option for respiratory support, but emerging options include diaphragm pacing and NMES, which provide advanced treatment options for patients. PEG tube placement can provide appropriate nutritional intake and prevent aspiration risk; spasticity and pain would benefit from a combination of pharmacological and non-pharmacological approaches. Likewise, high-frequency ultrasound-guided Botox for sialorrhea and BCIs for communication will be revolutionizing the ALS care model. Especially when we can couple personalized understanding of symptom control with multidisciplinary approaches, biomarker monitoring, and technology advances, we can improve symptom control, quality of life, and survival in rapid ALS progressors [341].

## 6. Emerging and Experimental Therapies in Clinical Trials: Pioneering Advances to Combat Fast ALS Progression

Investigational therapies highlight ALS mechanisms associated with rapid disease progression: neuroinflammation, mitochondrial dysfunction, protein-mediated toxicity, and aberrant RNA metabolism. For fast progressors, accelerated motor decline and increased physiological stress provide an argument for biomarker-guided strategies (for example, NfL, neurophysiology) and a careful approach to reporting drug and clinical signals as indicators of change. Below we summarize the approaches within trials demonstrating the intersection with fast progression biology [342].

### 6.1. CNM-Au8: A Bioenergetic Catalyst for Restoring Mitochondrial Function and Cellular Energy

Mitochondrial dysfunction and depletion of ATP are central features in ALS and likely exacerbate fast progressors. CNM-Au8 (catalytically active gold nanocrystals) restore bioenergetic homeostasis whereby the nanoparticle facilitates redox reactions, creating more ATP, fostering oxidative injury reduction and stabilization of a dysfunctional mitochondrial membrane [343].

Preclinical work implicated axonal support/uptake in delayed disease onset; the REPAIR-ALS Phase 2 trial is evaluating measures of neurophysiology (e.g., quality of MUs, MUNIX, and NfL). Combination regimens and biomarker-guided dosing are only hypothesis generating and informed predominantly by markers of mitochondrial stress (e.g., ATP, ROS, membrane potential) [252,344].

### 6.2. Verdiperstat: Targeting Microglial Overactivation and Neuroinflammation

Microglial activation mediates ongoing oxidative and inflammatory injury in ALS. Verdiperstat (AZD3241), an irreversible myeloperoxidase (MPO) inhibitor, is thought to reduce both hypochlorous acid and other related reactive intermediates [345].

Preclinical studies have indicated reduced neuroinflammation and axonal damage; HEALEY ALS Platform Trial (Regimen B) clinical outcomes include ALSFRS-R, inflammatory biomarkers, and survival [249]. The table below (Table 2) is intended to identify key biomarkers utilized in ALS clinical trials, their pathophysiologic significance, and clinical relevance for personalized treatment strategies.

Emerging combination strategies involve pairing verdiperstat with anti-inflammatory agents, such as IL-6 blockers (e.g., tocilizumab) or TNF-α inhibitors (e.g., XPro1595), to provide a dual anti-inflammatory approach. Additionally, cotreatment with mitochondrial-targeting therapies like CNM-Au8 or SS-31 could provide synergistic benefits by reducing both inflammation-induced and metabolic-stress-driven neurodegeneration. Machine learning algorithms integrating multiomics data (e.g., cytokine profiles, genetic predisposition markers, and NfL levels) are being used to predict which patient subgroups would respond most favorably to verdiperstat-based combination regimens [359].

### 6.3. Pridopidine: Sigma-1 Receptor (S1R) Modulation for Neuroprotection and Motor Function Recovery

The sigma-1 receptor (S1R) at the endoplasmic reticulum–mitochondria interface regulates calcium signaling processes, proteostasis, and responses to stress. Pridopidine, a selective S1R modulator, aims to stabilize interorganelle communication and mitigate cellular stress [360].

Preclinically, data indicates improved axonal resilience and neurotrophic support (BDNF, GDNF) [361]. In the HEALEY Platform (Regimen F), assessment scales include ALSFRS-R and biomarkers of synaptic or axonal integrity, with overall outcome assessments and some preliminary combination studies exploring additional neurotrophic approaches [362,363].

### 6.4. Wave ASOs: RNA-Based Therapeutics Targeting C9orf72-Related ALS

C9orf72 repeat expansions are known to generate toxic RNA foci, as well as dipeptide repeat proteins (DPRs) affecting RNA metabolism and nucleocytoplasmic transport. Stereopure Wave ASOs are designed to lower the levels of mutant transcripts while preserving the expression of the wild-type [364].

Limited early work has documented decreased CSF DPR species, as well as promoting some favorable signals in neurophysiology or axonal injury markers (e.g., NfL); optimization is ongoing in terms of delivery strategies (lipid nanoparticles, AAV) and regimen personalization based on expansion size, methylation status and biomarkers [365,366].

### 6.5. ALS iPSC-Based Therapy: Personalized Stem Cell Replacement and Genetic Correction

Induced pluripotent stem cell (iPSC) approaches offer the opportunity to produce patient-specific neural/glial derivatives, and potentially a CRISPR-Cas corrected one, prior to differentiating [367].

Preclinical studies have indicated trophic support and improved proteostasis (e.g., engineered astrocytes that specifically targeted TDP-43/SOD1 aggregates); currently, biomaterial scaffolds are being studied and tested to support graft stability. The CLINE-ALS trial is investigating long-term safety and feasibility using imaging and physiologic monitoring [368,369].

These investigational strategies should offer an overall mechanism by which the mechanisms may be accelerated in fast progressors. In our opinion trials should: (1) enrich participants based on high-risk biology (elevated baseline NfL and/or genotype, etc); (2) prespecify the pharmacodynamic thresholds; and (3) treat biomarker improvement as supportive unless it parallels functional improvement. We purposely do not include exhaustive drug catalogues and unsupported combinations to avoid overwhelming situations [21,370].

## 7. Clinical Trials for Fast Progressors: Precision Research in the Era of Personalized ALS Therapies

Clinical trials are the linchpin of any new research program in ALS and provide opportunities for fast progressors to trial precision, biomarker-guided treatment approaches. The fast progressor subset of individuals diagnosed with ALS have unique and significant motor decline driven by genetic loading, metabolic stress, and neuroinflammation. Trials are more often embedding genetic screening and multiomics and prospective biomarker readout data to enrich study cohorts and inform decision making. We will reflect on a few relevant examples and design principles that are relevant to fast progressor patients.

### 7.1. TRICALS: A European Powerhouse for ALS Clinical Trials and Innovation

TRICALS operates a multinational network with biomarker-derived stratification and individualized treatment paradigms [371].

The flagship example is ATLAS (NCT04856982) [372]; this is a presymptomatic SOD1 intervention utilizing longitudinal NfL to initiate tofersen therapy before clinical onset, demonstrating early, biomarker-guided treatment in high-risk carriers with limited clinical expression [59].

The TRICALS network also collaborates with ENCALS and initiates data science approaches to refine study protocols for enrichment and dosing considerations for attendees in anticipated rapid decline [373].

### 7.2. European Union Clinical Trials Register: Comprehensive Access to ALS-Specific Trials

The EU Clinical Trials Register collects randomized interventional studies and any of the mechanisms reviewed that may be pertinent to fast progressors (neuroinflammation, mitochondrial dysfunction, toxic protein deposition, axonal degeneration) [374].

As a recent example, ALS-COVERT (NCT04972487) is a tau-directed strategy to measure peripheral axonal stability and also studies exploring EphA4 inhibition for motor neuron viability and bioenergetic/ER stress approaches in a combination approach [263]. In presenting a few representative contexts we are not attempting to be exhaustive or summarize drugs for clarity.

### 7.3. ALS Therapy Development Institute (ALS TDI) and the ALS Trial Navigator

ALS TDI has a well-established role in accelerating ALS therapeutics and matching participants to studies based on biomarker and genetic profiles. For example, AT-1501 (Phase 2) provides a therapeutic targeting CD40L, with the aim of modifying inflammation. The earlier works for this agent support a biological rationale and it appears to be compatible with a number of standard agents, with definitive clinical benefit to be determined [375,376].

### 7.4. Global Trials: Asia-Pacific and Australia’s Contributions to ALS Research

Programs in the Asia-Pacific region and Australia provide expanded opportunities and resources for both genotype diversity and therapies. The list of promising areas of activity is too expansive to consider, but examples from Japan may include mesenchymal stem cell approaches with trophic support; national, engineering-level screenings of engineered astrocytic approaches in South Korea; and NTCELL (encapsulated choroid plexus) in Australia, for use in sustained delivery of neurotrophic/anti-inflammatory treatment [284,377].

### 7.5. Tofersen for Presymptomatic SOD1 ALS (ATLAS Trial): Pre-Emptive Neuroprotection

ATLAS (NCT04856982) serves to provide a proof of concept for an early intervention threshold for presymptomatic SOD1-mediated neurodegeneration. Utilization of tofersen has shown targeting of SOD1 mRNA (and subsequent apparent reduces mutant SOD1) to limit downstream insults, and the anticipated clinical benefit of this prophylactic treatment is provided in a prospective manner [372].

### 7.6. AI-Powered Adaptive Trial Designs: Revolutionizing ALS Research

Model-based, biomarker-informed adaptive designs enable response-adaptive randomization, dose adjustments, arm switching, and/or early stopping as a function of interim signals (e.g., NfL/physiologic changes), which could be advantageous for fast progressors [378,379].

Trials for fast progressors should (i) enrich for high-risk biology (high or elevated baseline NfL and/or genotype); (ii) provide prespecified relevant thresholds for pharmacodynamics but focus on functional targets; and (iii) be prepared to utilize innovative adaptive features if the biomarker objectives are not met. The emphasis is on design principles, and we hope to provide only a few representative programs instead of a long point form list while retaining focus and readability [26].

## 8. Conclusions: A Paradigm Shift in the Treatment of Fast ALS Progressors

Fast-progressing ALS is one of the most devastating forms of neurodegenerative disease. It manifests itself in clinical features of aggressive motor decline, sudden metabolic collapse, and persistent neuroinflammation. Ordinary treatment protocols for fast progressors have been ineffective, and for many decades clinician choices have been totally remedial and palliative. Luckily, this is changing. The data illustrate that simply using some combination of the multicomplex modes of treatment or therapeutic concepts, using gene-based interventions, and/or using neuroprotective therapeutics aiming at each stage of the disease concept—starting with the genetic genesis leading to the cellular downstream pathways—can create tremendously high value. As we have discussed, it is possible that the reality of fast ALS progression is not a reality but rather a daunting challenge. The interaction of biologic thesis, treatment innovation, personalized medicine, and worldwide outreach is creating a future for ALS where it can be recast from a fatal disease to a disease of chronic management. This is not an end but rather the beginning of a revolution in treatment for ALS.

### 8.1. Summary of Key Findings: Addressing the Complex Pathophysiology of Fast Progressors

Fast-progressing ALS represents a population of unique patients. Fast progression of disease has many factors attributed to the patients’ accelerated decline, derived from the mix of genetics, metabolic function, and neuroinflammation. In our review we have discussed some of the complexities of some of the biggest drivers of fast progressions—mitochondrial dysfunction, neuroinflammation, toxic protein load, and genetics. Mitochondrial dysfunction diminishes the available ATP for motor neurons and creates oxidative and cellular injury. At the same time, activated astrocytes and microglia produce cytokines, which catalyze the common neuroinflammatory cascade and lead to axonal degeneration. Potentially, toxic protein load (i.e., loading of TDP-43, SOD1, and dipeptide repeat proteins by C9orf72 ALS) produces another form of cellular load. People with fast-progressing genetic variants (e.g., SOD1, C9orf72, FUS, or TARDBP) will amplify the above-listed mechanisms, accelerating the disease itself, while reducing opportunities due to time constraints.

There are new treatment options emerging that target those mechanisms directly. There is evidence, for example, with gene therapies like tofersen for SOD1 and Wave ASOs for C9orf72 expansions demonstrating reductions in the levels of mutant protein in humans and positive changes in the clinical course. One bioenergetic nanocatalyst that has shown an ability to restore cellular energy after disruption is called CNM-Au8, in addition to a mitochondrial and ER stress modulator called AMX0035, showing protective abilities for motor neurons. Other trials are using iPSCs for end-stage therapy (motor neuron replacement) and glial cell supportive approaches to provide a sustainable tendency for regeneration. In addition to the mechanisms listed above, these new treatments can also be calibrated with clinical intervention approaches that include real-time biomarker monitoring, adaptive clinical research trials, and personalized dosing strategies. We are not locked into managing symptoms with medications. We can treat fast-progressing ALS throughout the entire trajectory of the disease.

### 8.2. The Critical Role of Early Identification and Biomarker-Driven Classification

Unfortunately, for fast-progressing ALS, treatment options are limited and therefore fast ALS identifiers and biomarker-driven classification are critical to initiate treatment in a timely manner. We have seen in the past clinical symptoms and suddenly assumed some form of constructive future neurodegeneration and damage, thus wasting precious time for fast ALS progressors. Fast ALS progressors were actually suffering more harm by our historical clinical symptom diagnoses because they were not starting treatment and purposely waited to receive treatment on receiving a diagnosis or in order to meet the so-called “diagnostic criteria” which delayed time to treatment when treatment could have substantiated change in the course of damage. The specific metrics of biomarkers using some bipolar assessment method would at least indicate what the clinician should see as the clinical problem through a whole further lens than just simply either treatment or no treatment. We saw biomarker changes in ways we could note months before we saw similar changes in patients’ conditions in terms of clinical symptoms. The prime example of this type of marker is NfL. To that end, NfL is one of the more commonly portrayed axonal damage markers, and it was obvious that medical treatment of neurodegeneration was needed long before the more typical clinical symptoms of observable damage that we see or document occurred. This we believe has been demonstrated by the observation of NfL in presymptomatic SOD1 mutation carriers and from the ATLAS trial. The bottom line is that we were able to show through mapping biomarkers that targeted and tracked biomarkers to target early intervention might delay the disease and possibly somewhat alter changes already present.

Lastly, biomarkers have been part of targeted therapies since they identify specific treatments that would be appropriate to patients receiving therapy. Patients would be matched to clinical trials based on their genetic defect or use of a targeted therapy that would have downstream consequences, which required identifying SOD1, C9-orf72, FUS, or TARDBP variants to yield the best placement of a patient in the treatment. Moreover, classifications of patients can be further speculated based on the molecular basis of the disease which may be driven through multiomics data (genomics, transcriptomics, and proteomics) with no further involvement from the patient on mechanistic pathways of their disease. Therefore, although biomarker-identified classifications provide a case for a treatment plan to be customized to the genetic or biological nature of the group of patients and add the patients to the treatments, which could potentially be more effective or potentially reduce side effects, if applicable. In addition, biomarkers have the potential to identify alternatives in mimicking and customizing ongoing treatment reactively during the course of treatment, using the biomarker response as a further informal accommodation mechanism to aid the psychosocial journey during the treatment of the illness.

### 8.3. Future Research Directions: The Road Ahead in Gene Therapy, Neuroprotection, and Clinical Innovation

The ALS research agenda will be dominated by multimodal therapies aimed at targeting disease onset and coprotecting against disease downstream impacts. Gene therapy has promise, as do new developments in ASOs, CRISPR-based recombinant editing, and vector-based gene delivery. There is a rationale to expand efforts towards a gene-targeted approach towards other variants based not only on the observed reductions in target protein in the clinic but also the resulting improvements in patient outcome, as seen in tofersen (SOD1). Ongoing studies with Wave Life Sciences’ C9orf72-targeting ASOs are addressing how RNA-based therapies can prevent dipeptide repeat protein toxicities and implicate aberrations in nucleocytoplasmic transport.

Neuroprotective compounds such as CNM-Au8 show potential to restore cellular homeostasis. CNM-Au8 aims to override mitochondrial dysfunction by replenishing ATP production and improving oxidative stress. The potential to combine mitochondrial stabilizers with autophagy inducers (i.e., trehalose or rapamycin) and anti-inflammatories (e.g., verdiperstat and tocilizumab) may provide a multitarget approach to slow disease with a cumulative pathway advantage.

Developments in AI and machine learning will further enhance combination therapy by accelerating trial designs, defining the ideal subgroup of patients, and predicting which treatment combination from available regimen options will generate the most impact based on multiomics data (genomics, transcriptomics, proteomics, and metabolomics) to select the most optimal drug combinations. Multiarm adaptive Phase 2 clinical trials that work to concurrently test several therapy combinations will reduce timelines for clinical approval and expedite the patients’ pathway to clinical validation and regulatory approval. Future ALS research will be defined by precision medicine, adaptability, and collaborative effort to ensure patients receive the appropriate treatment, at the appropriate time.

### 8.4. A Call for Personalized Medicine and Global Collaboration

Customizing treatment to the individual patient is not a vision for the future—for ALS fast progressors it is urgent. No two people’s experiences of ALS are the same and there exists significant genetic and clinical heterogeneity, which necessitates customizing treatment for the individual patient. Multimodal treatment approaches that include gene therapies, neuroprotectants, and agents that modulate autophagy will need to be personalized based on genetic testing, biomarker consideration, and ongoing assessment. The integration of wearable devices as well as digital biomarkers will enhance the ability for our patients and clinical researchers to monitor disease progression, while working towards adjusting treatment options in real time.

Global collaboration is essential, and initiatives like the TRICALS, the HEALEY Platform Trial, and the ALS Therapy Development Institute are proving that the power of international collaboration can accelerate the development of drugs and trials. Global data-sharing networks and coordinated regulatory pathways can help the scientific community ensure every breakthrough in one region can, at lightning speed, be accessed by patients across regions worldwide. Since ALS is truly a worldwide challenge, we can only overcome it through worldwide solutions.

### 8.5. The Urgency to Act: A Window We Cannot Afford to Miss

Time is the most precious commodity for ALS patients who experience fast disease progression. For every day that passes without effective treatment, irreversible harm occurs. The scientific community must act, in an urgent and precise manner, to uncover solutions and fast-track the translation of experimental therapies from laboratory to clinic. We must develop fast-track regulatory pathways, wider access to experimental therapies, and adaptive trial design focused on early intervention in order to deliver life-altering therapies within the limited window of opportunity.

But even urgency will not be enough. We will need to be strategic. Through the combined use of biomarker-driven patient stratification, AI-driven trial optimization, and personalized dosing regimens, we can make sure each patient receives the best and most effective combination of therapies based on their unique disease profile. The resources are available to make this happen; we need to strategically implement these tools at scale.

### 8.6. Final Words: Transforming ALS from Fatal to Treatable

ALS has taken too many lives and devastated too many families, but we are at a moment of change. With new approaches to gene therapies, neuroprotective agents, autophagy inducers, and monitoring technology, a new era of possibility is around the corner when we may be able to stop progression and perhaps even reverse it. Through continued international collaboration, individualized treatment approaches, and innovation, we can see that ALS is on its way to transforming from a terminal diagnosis to a manageable chronic condition.

These pages should inspire reflection of how far we have already come (with regard to developing experimental therapies for ALS) and galvanize our commitment to what is left to do. The battle to overcome ALS is far from over, but for the first time ever we can imagine victory within reach. For as long as we press on with our innovations, working collaboratively on a global scale and with urgency, we will change the lives of ALS patients, one discovery at a time.

## Figures and Tables

**Figure 1 ijms-26-08072-f001:**
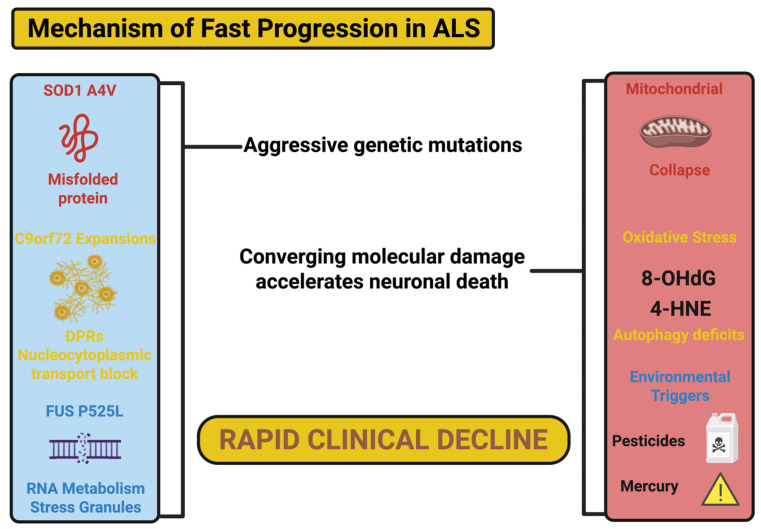
Mechanistic framework summarizing the drivers of fast progression in ALS. This schematic aims to provide an integrative overview of the key factors believed to underlie the rapid progression observed in certain ALS phenotypes. The left panel outlines representative genetic variants—SOD1 A4V, C9orf72 expansions, and FUS P525L—that have been associated with toxic protein aggregation, altered nucleocytoplasmic transport, and impaired RNA metabolism. These alterations are proposed to initiate a cascade of molecular disturbances, including mitochondrial dysfunction, oxidative stress (e.g., increased 8-OHdG and 4-HNE), and deficits in autophagy. The figure also intends to highlight the possible contribution of environmental factors, such as exposure to mercury and pesticides, which may further exacerbate neuronal vulnerability. Together, these converging processes are thought to accelerate motor neuron degeneration and contribute to the clinical features associated with fast ALS progression.

**Figure 2 ijms-26-08072-f002:**
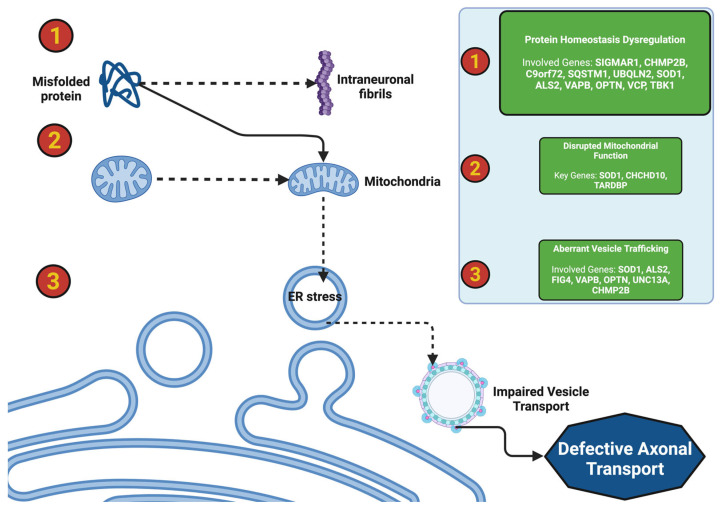
Interconnected pathological mechanisms driving ALS fast progression. Misfolded proteins aggregate into intraneuronal fibrils, inducing mitochondrial dysfunction, ER stress, and impaired vesicle trafficking. These disruptions lead to defective axonal transport and motor neuron degeneration. Key genetic variants (e.g., SOD1, C9orf72, FUS, TARDBP) contribute to the dysregulation of protein homeostasis, mitochondrial function, and vesicle trafficking, highlighting therapeutic targets for multimodal interventions.

**Figure 3 ijms-26-08072-f003:**
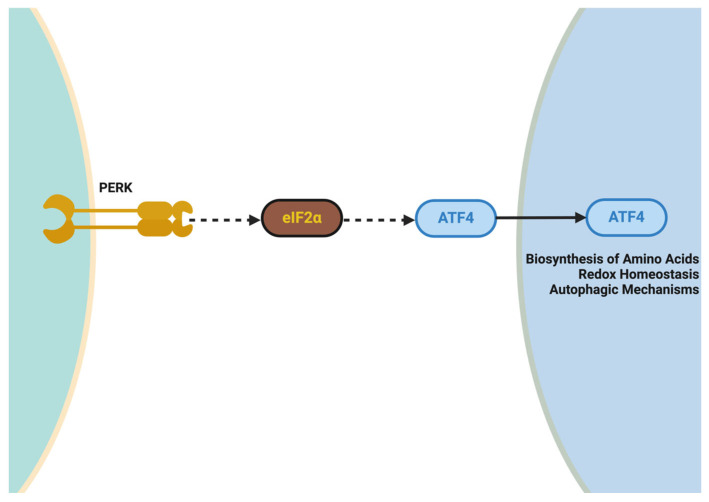
The PERK-eIF2α-ATF4 pathway and its downstream effects. PERK activation leads to eIF2α phosphorylation, which triggers ATF4-mediated regulation of processes including amino acid biosynthesis, redox homeostasis, and autophagy. Dysregulation of this pathway under chronic ER stress exacerbates neurodegeneration in ALS fast progressors, highlighting its importance as a therapeutic target.

**Table 1 ijms-26-08072-t001:** This table categorizes critical ALS pathways, the genetic variants or dysregulated processes associated with fast progression, and the corresponding therapeutic interventions.

Key Pathway/Process	Genetic Variants/Targets	Pathophysiological Mechanism	Therapeutic Strategies	References
Protein misfolding and aggregation	SOD1 (A4V, G93A variants)	Misfolded SOD1 aggregates induce oxidative stress, impair proteostasis, and disrupt mitochondrial function.	ASOs (tofersen), arimoclomol (HSP induction), SS-31 (mitochondrial stabilizer)	[17,245]
RNA dysregulation	C9orf72 expansions	Formation of toxic RNA foci and dipeptide repeat proteins (DPRs) disrupt nucleocytoplasmic transport and protein homeostasis.	Wave ASOs, CRISPR-based genome editing, trehalose (autophagy inducer)	[246,247,248]
Neuroinflammation	Immune activation (IL-6, TNF-α)	Chronic microglial activation releases ROS and pro-inflammatory cytokines, contributing to axonal degeneration and cell death.	Tocilizumab (IL-6 blocker), verdiperstat (MPO inhibitor), XPro1595 (TNF-α modulator)	[249,250,251,252]
Mitochondrial dysfunction	PGC-1α, SOD1 variants	Impaired oxidative phosphorylation and calcium overload induce ATP depletion and apoptosis.	CNM-Au8 (bioenergetic enhancer), AMX0035 (ER-mitochondrial stabilizer), SS-31 (ROS reduction)	[253,254,255]
Defective autophagy	FUS, TARDBP (TDP-43)	Autophagy failure leads to accumulation of protein aggregates, impaired clearance of damaged organelles, and synaptic loss.	Trehalose (autophagy enhancer), rapamycin (mTOR inhibitor), autophagy lysosome enhancers (TFEB activation)	[256,257,258]
Nucleocytoplasmic transport	C9orf72, FUS variants	Defects in nuclear pore complexes (NPCs) impair transport of proteins and RNAs, leading to genomic instability and toxicity.	Small molecules targeting NPC stabilization, ASOs for RNA clearance	[259,260]
Oxidative stress	SOD1, ATXN2 expansions	Increased ROS production damages lipids, proteins, and DNA, further exacerbating mitochondrial collapse.	Edaravone (antioxidant), AMX0035, SS-31, MitoQ	[261,262]

**Table 2 ijms-26-08072-t002:** The table highlights critical biomarkers currently used in clinical trials to guide early diagnosis, predict disease progression, and assess therapeutic response in ALS fast progressors. These biomarkers reflect key pathological processes, including axonal degeneration (NfL), neuroinflammation (IL-6, TNF-α), protein aggregation (pTDP-43), mitochondrial dysfunction, and toxic RNA misprocessing.

Biomarker Type	Specific Biomarker	Source	Pathophysiological Relevance	Clinical Application	References
Neurofilament biomarkers	Neurofilament light chain (NfL)	Cerebrospinal fluid (CSF), plasma	Marker of axonal injury; elevated during active neurodegeneration	Prognostic marker for disease progression, therapeutic monitoring in trials (e.g., tofersen, Wave ASOs)	[147,346]
Inflammatory biomarkers	IL-6, TNF-α	Plasma, CSF	Elevated in neuroinflammation due to activated microglia and astrocytes	Predicts neuroinflammatory burden; informs use of anti-inflammatory therapies like verdiperstat and XPro1595	[347,348]
Protein aggregation markers	Phosphorylated TDP-43 (pTDP-43)	CSF	Reflects cytoplasmic mislocalization and aggregation, hallmarks of ALS pathology	Diagnostic marker for ALS and disease subtyping; potential for tracking responses to autophagy-based treatments	[349,350]
Mitochondrial biomarkers	Lactate, pyruvate ratio	Plasma, CSF	Indicates mitochondrial dysfunction and impaired oxidative phosphorylation, common in ALS fast progressors	Used to assess metabolic interventions targeting mitochondrial function, e.g., CNM-Au8, AMX0035	[351,352]
RNA biomarkers	C9orf72 RNA foci	Neuronal tissues, blood	Toxic RNA aggregates formed due to hexanucleotide expansions, disrupting nucleocytoplasmic transport	Diagnostic and therapeutic monitoring in familial ALS cases with C9orf72 expansions; informs ASO or CRISPR-based therapies	[243,353]
Genetic biomarkers	SOD1, FUS, TARDBP variants	Blood, saliva, CSF	Identify genetic subtypes prone to faster progression due to toxic gain-of-function variants	Used for personalized therapeutic selection; informs trial eligibility for gene-based therapies (e.g., tofersen for SOD1)	[146,354]
Imaging biomarkers	Brain MRI (corticospinal tract)	Structural MRI	Detects upper motor neuron degeneration and corticospinal tract atrophy	Used for early diagnosis, disease staging, and tracking motor neuron degeneration over time	[355,356]
Functional biomarkers	Motor unit number estimation (MUNIX)	Electrophysiological recordings (EMG)	Reflects motor neuron functional integrity and axonal degeneration	Assesses disease progression and response to therapies targeting axonal preservation	[357,358]

## Data Availability

Not applicable.

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
