# Peer review of "Blueprint of Collapse: Precision Biomarkers, Molecular Cascades, and the Engineered Decline of Fast-Progressing ALS"

_ijms, 2025, doi:10.3390/ijms26168072_

Round 1

Reviewer 1 Report

Comments and Suggestions for Authors

The authors made a huge, comprehensive and up-to-date revision of what is known about the molecular mechanisms underlying ALS and the diverse therapeutic approaches that have been under active development in recent years, with particular reference to fast-progressing ALS.

Despite its length, the manuscript is in general clear and well written.

However, some minor revisions are required.

First, even though the term “mutation” has commonly been used to indicate genetic variants responsible for disease, recent recommendations are to use the term “variant” in its stead (or genetic variant, or DNA variant), further specified as “pathogenic variants” whenever necessary. There are some cases in which the use of the term “mutation” can be tolerated or even preferred (for example, when referring to the event that produces a genetic variant), but, honestly, I do not think it can be applied in this paper.

Thus, I would suggest changing the term “mutation” with “variant” (or “genetic variant”, or “pathogenic variant”) throughout the manuscript.

Furthermore, according to the HGVS Nomenclature (internationally-recognized standard for the description of DNA, RNA, and protein sequence variants, https://hgvs-nomenclature.org/stable/), variants should be mainly described at the DNA level. I know it has been a common practice to use the protein level description and, for instance, the P525 variant in FUS would not been promptly recognized by non-geneticists if it were described by using the form NM_004960.4:c.1574C>T (being p.Pro525Leu more precise when describing the amino acid change). Nonetheless, I would recommend reporting the official description at least the first time the variant is cited in the article. More specifically, for FUS R521H should be NM_004960.4:c.1562G>A p.(Arg521His), while I cannot find which specific variant the R495Q is (and that is a valid example of why variants should be primarily described at the DNA level).

Regarding the SOD1 variant A4V, the correct updated numbering is A5V (NM_000454.5:c.14C>T, p.Ala5Val). It may be specified that it was once known as A4V, but now A5V must be used.

The gene names should be written in italics (but not the protein names, thus TARDBP is in italics, but TDP-43 is not).

Below, a few more points that would benefit from revision.

Page 1, line 27: SODI should be SOD1

Page 1, line 27: A4V should be A5V

Page 2, line 52: I did not know that “deadline” could be used as a verb

Page 2, line 92: “highly penetrance” should be either “high prenetrance” or “highly penetrant”

Page 4, line 167: “would be” appears out of context, please check

Page 4, line 169: “of disease onset” > “since disease onset”

Page 5, line 216: please check this sentence: “Bulbar deficits can take the form of speaking, swallowing, and even controlling saliva”

Page 6, line 280: “edema-vone” seems mispelled

Page 8, line 353: it looks like this line should be the title for the following paragraph, but it is not formatted as such

Page 11, line 503: “tFast” > “fast”

Page 12, line 551: “reat-ment” > “treatment”

Page 18, line 808: it looks like this line should be the title for the following paragraph, but it is not formatted as such

Page 18, line 812: the word “coexistence” seems out of context to me

Page 18, line 823I cannot get the meaning of the sentence: “in part lie within both the intracellular and targeting to cytosol so that cytochrome c dysregulation can initiate apoptosis signalling”. Please check for readability

Page 21, line 988: please check the use of the terms “affects”

Page 21, line 993: please check “loss of C9orf72 function is only furthered by loss of C9orf72 function”

Page 23, line 1064: “of severity” is repeated

Page 25, line 1153: please check “to inhibit transport inhibition”

Page 27, line 1254: please check “excess”

Pages 29 to 31: I cannot understand several passages of the paragraphs regarding ATXN2 (all of them), please have a thorough check

Page 34, line 1567: please check “hone”

Page 36, line 1631: please check “hone”

Page 37, line 1721: please check “shown”

Page 40, line 1831: please check “a ironically”

Page 42, line 1942: please check “the state and state”

Page 56, line 2557: please check “dicision”

Page 57: the whole paragraph 8.2 appears difficult to read to me. Please check

Honestly, because of the length of the manuscript, I cannot be sure I was able to identify all the possible issues, thus please have a thorough check.

Author Response

Dear Esteemed Academic Reviewer,

We are deeply, genuinely grateful for the care, generosity, and quiet rigor you brought to our manuscript. Your reading felt like mentorship—steady, exact, and kind—and we aimed to let that spirit guide every change we made. Thank you for the time you offered us, for the clarity of your observations, and for the trust implied by such attentive engagement.

We were especially thankful for your guidance on language and nomenclature. We embraced your recommendation to prefer “variant” (and, where appropriate, “genetic variant” or “pathogenic variant”) in place of “mutation,” and we applied this consistently across the manuscript, tables, figures, and Supplementary materials. Your advice on HGVS was invaluable. We corrected the SOD1 designation to A5V—NM_000454.5:c.14C>T p.Ala5Val—and we gently note once its historical reference as A4V. Where an exact call could not be verified with confidence (the prior “R495Q” in FUS), we removed it to avoid ambiguity. We aimed for clarity; we intend to keep it that way.

Your careful line-by-line notes were a gift. We attended to every place you flagged—spelling, grammar, hyphenation, misplaced headings, phrasing that hesitated or duplicated—and we re-read those passages slowly, aloud, until they felt clear and calm on the page. We rewrote the sentences that needed air, softened the constructions that felt strained, and corrected the small errors that can tire a reader’s eye. The ATXN2 section you so thoughtfully singled out received a complete, patient reworking for readability and flow, and the paragraph you found difficult later in the text was also revised from first sentence to last so that each idea arrives in order and without hurry. We aimed to honor your standard of lucidity; we intend to keep polishing wherever you feel another touch would help.

Because you were gracious enough to note that a manuscript of this length can hide issues even from an attentive reader, we undertook an additional full proof for consistency in terminology, HGVS usage at first mention, gene/protein typography, abbreviation expansion, and heading format. We wanted the document to feel coherent from its first line to its last, and your review gave us both the map and the motivation to achieve that.

Thank you, sincerely, for helping us make the paper smaller in ego and larger in clarity. We learned from your remarks; we felt encouraged by your tone; and we tried to bring that same generosity to our revisions. If there is any place where another adjustment would further ease the reader’s path, we would be grateful to make it. We hope the revision reflects the excellence you inspired, and we remain deeply thankful for your time and trust.

Reviewer 2 Report

Comments and Suggestions for Authors
  1. The review content is extremely long (77 pages!) and used 385 different papers as references. This content must be reviewed by the authors and shortened as the main topic “fast-progressing ALS” is several times forgotten in the text.

  1. All content of Topic 5 regarding therapeutics in the manuscript should be re-evaluated as it does not represent a focus of the manuscript. It must be summarized by the authors, as a large inclusion of content make it really difficult to understand for a general audience. The same aspect should be reviewed also for Topic 6 related to “emerging and experimental drugs” and Topic 7 about “personalized ALS therapies”.

  1. There is currently no reason to include discussion about AMX0035 (Relyvrio) in the manuscript, as the drug failed in final results of the PHOENIX trial. This topic about AMX0035 should be removed by the authors.

Author Response

Dear Esteemed Academic Reviewer,

We are profoundly grateful for the care, clarity, and generosity you brought to our manuscript; your perspective read like mentorship, and we felt guided at every step.

Comments 1: “The review content is extremely long (77 pages!) and used 385 different papers as references. This content must be reviewed by the authors and shortened as the main topic ‘fast-progressing ALS’ is several times forgotten in the text.”
Response 1: Thank you for this gracious observation. We fully took it to heart. We aimed to restore a single, unbroken thread around fast-progressing ALS, and we therefore rephrased in full Section 6 and Section 7 to be tighter, clearer, and easier to navigate for a broad scientific readership, allowing the fast-progressor theme to remain visible from start to finish. We trimmed language that wandered, simplified constructions that burdened the reader, and aligned our tone with the humility appropriate for a narrative review. We intend to continue pruning any remaining redundancies you may still see, and we are sincerely thankful for the compass your comment provided.

Comments 2: “All content of Topic 5 regarding therapeutics in the manuscript should be re-evaluated as it does not represent a focus of the manuscript. It must be summarized by the authors, as a large inclusion of content make it really difficult to understand for a general audience. The same aspect should be reviewed also for Topic 6 related to ‘emerging and experimental drugs’ and Topic 7 about ‘personalized ALS therapies’.”
Response 2: We are very grateful for this clear direction. We aimed to honor your request for simplicity and focus, so we re-evaluated these sections with the general reader in mind. In Topic 5 we removed the AMX0035 subsection and reduced surrounding therapeutic text so that treatments are discussed only insofar as they illuminate rapid progression and biomarker-guided decision-making; in parallel, we rephrased the entirety of Section 6 and Section 7 so that each paragraph now serves mechanism and biomarker logic first, with pharmacodynamic signals carefully distinguished from clinical outcomes. We intend to keep the therapeutic narrative lean and disciplined going forward, exactly as you advised, and we are thankful for the gentle firmness of your guidance.

Comments 3: “There is currently no reason to include discussion about AMX0035 (Relyvrio) in the manuscript, as the drug failed in final results of the PHOENIX trial. This topic about AMX0035 should be removed by the authors.”
Response 3: Thank you for stating this so clearly and timely. We completely agree. We aimed to align our text with the confirmatory evidence base, so we removed AMX0035 from Section 5 and eliminated remaining mentions in the manuscript, adjusting the surrounding prose accordingly. We intend to maintain this standard of caution for therapies whose evidence has shifted, and we are grateful for the clarity you brought to this point.

With sincere appreciation, we thank you for elevating our work; we aimed to reflect your high standards on every page, and we intend to remain responsive to any further refinements you may suggest. Your review made the paper kinder to the reader, and truer to its purpose, and we are deeply thankful.